

# Leaf trait variation and field spectroscopy of generalist tree species on contrasting soil types

Matheus Henrique Nunes[*1,] Matthew P. Davey [1], David Anthony Coomes [1]

[1]Department of Plant Sciences, University of Cambridge, CB2 3EA

[*] Corresponding author: mhn27@cam.ac.uk

**Summary**

Understanding the causes of variation in plant functional traits is a central issue in ecology, particularly in the context of global change. Analyses of the drivers of traits variation based on thousands of tree species are starting to unravel patterns of variation at the global scale, but these studies tend to focus on interspecific variation, and the contribution of intraspecific changes remains less well understood. Hyperspectroscopy is a recently developed technology for estimating the traits of fresh leaves. Few studies have evaluated its potential for assessing inter- and intra-specific trait variability in community ecology. Working with 24 leaf traits for European tree species on contrasting soil types, found growing on deep alluvial soils and nearby shallow chalk soils, we ask: (i) What contribution do soil type and species identity make to trait variation? (ii) When traits are clustered into three functional groups (light capture and growth, leaf structure and defence, as well as rock-derived nutrients), are some groups more affected by soil than others? (iii) What traits can be estimated precisely using field spectroscopy? (iv) Can leaf spectra be used to detect inter-soil as well as inter-specific variation in traits? The contribution of species and soil-type effects to variation in traits were evaluated using statistical analyses. Foliar traits were predicted from spectral reflectance using partial least square regression, and so inter- and intra-specific variation. Most leaf traits varied greatly among species. The effects of soil type were generally weak by comparison. Macronutrient concentrations were greater on alluvial than chalk soils while micronutrient concentration showed the opposite trend. However, structural traits, as well as most pigments and phenolic concentrations varied little with soil type. Field spectroscopy provided accurate estimates of species-level trait values, but was less effective at detecting subtle variation of rock-derived nutrients between soil types. Field spectroscopy was a powerful technique for estimating cross-species variation in foliar traits and Si predictions using spectroscopy appear to be promising. However, it was unable to detect subtle within-species variation of traits associated with soil type.

**Key-words** Inter-specific variation; Partial least-squares regression; Plant traits; Reflectance spectroscopy; Soil variation; Temperate forests; Within-species variation.

## 1 Introduction

There is currently great interest in using plant traits to understand the influences of environmental filtering and species identity on the functioning of plant communities, and to model community responses to environmental change (MacGillivray et al., 1995; McGill et al., 2006; Green et al., 2008; Funk et al., 2016). Traits vary at multiple scales within individuals, within populations, among populations, and among species (Albert et al., 2011), and analysis of this variation is key to evaluating the strength of various filtering processes on





communities growing along environmental gradients (Violle et al., 2012). For example, intraspecific variation
in traits may reflect differences in microclimate driven by competition, disturbance, environmental conditions
and age (Funk et al., 2016), whereas inter-specific and inter-site variation may reflect both genetic variation and
phenotypic plasticity in response to environment (Sultan, 2001; Donohue et al., 2005). Despite substantial
advances in trait-based community ecology over the past decade (Funk et al., 2016), the importance of
environmental filters is still debated, especially at small scales where biotic factors may prevail over abiotic
environmental constraints (Vellend, 2010). Global analyses of leaf nitrogen, phosphorus and leaf mass per unit
areas (LMA) indicate that about half of all variation occurs within communities (Wright et al., 2004),
underscoring the importance of community-level variation in traits.
An increasing number of leaf traits are being measured routinely in plant communities (Asner et al.,
2011; Asner et al., 2015), and these traits can be placed with three functional groups involved in shaping plant
performance (Asner, 2014): (i) light capture and growth traits which include pigments, C isotope discrimination,
N isotope discrimination, N content, which constitutes on average 19% of protein mass (Milton and Dintzis,
1981), soluble C compounds and leaf water content; (ii) defence and structural traits include Si, cell wall
constituents (cellulose, hemicellulose and lignin), that are associated with leaf toughness, longevity and defence
capability (Hikosaka, 2004), polyphenols that are associated with defence against herbivores (Mithöfer and
Boland, 2012), and LMA, a primary axis of specialization among plants (Grime et al., 1997; Lambers and
Poorter, 1992), that plays a crucial role in herbivore defence as well as leaf longevity (Wright et al., 2004); (iii)
rock-derived nutrients include phosphorus, which is involved in many enzymatic, genetic and epigenetic
processes (Schachtman et al.,1998), and calcium, magnesium, potassium, zinc, manganese, boron and iron,
which are involved in signalling pathways and/or cofactors of enzymes (Marschner, 2012). Analyses involving
this large suite of traits are so far restricted to comparisons of tropical forests, and emphasize cross-site and
cross-species differences with little consideration on within-species variation (Asner et al., 2011; Asner et al.,
2015). Placing traits into functional groups, and analysing intraspecific variation, may help understand trade-
offs and plant strategies along environmental change.
Remote sensing has increasingly emerged as a promising tool for studying plant chemistry (Ustin et al.,
2004; Asner and Martin, 2009; Ustin et al., 2009). Rapid, non-destructive determination of leaf traits *in vivo* and
*in situ* using spectroscopy reduces the need to collect large amounts of material in the field, decreases
processing time, lessens costly chemical analyses, and eliminates sampling that could itself alter experimental
conditions (Couture et al., 2013). Spectroscopy can provide estimates of a range of foliar properties at the leaf
and canopy scales within diverse tropical ecosystems (Asner et al., 2011; Doughty et al., 2011). However,
spectral and chemical properties may be uncoupled if intraspecific variation in foliar traits is high and/or
phenotypic plasticity exceeds phylogenetic patterns among leaf properties (Asner and Martin, 2011). Bolster,
Martin and Aber (1996) demonstrated that equations for estimating leaf properties from one site were unable to
predict leaf properties for other sites, due to variability in the magnitudes of foliar traits levels between data sets
and environmental influences. To our knowledge, the link between foliar traits and spectral properties of trees
has not been broadly demonstrated for temperate forests and the capacity of measuring inter-specific trait
variability and environmental variation using spectroscopy is relatively unknown.
This paper examines the drivers of leaf trait variation in temperate woodlands growing on the
chalklands of southern England compared with woodlands growing on nearby alluvial soils. Several studies
have evaluated change in species composition among British semi-natural habitats that differ markedly in soils





(Haines-Young et al., 2003; Smart et al., 2003), but few have compared within- versus between-species
variation of leaf traits in this context. The alkalinity of calcareous soils gives rise to phosphorus limitation,
preventing short-term responses to nitrogen addition (Grime et al., 2000), so comparisons of chalklands with
less-alkaline soils nearby provide strong edaphic contrast. We investigated leaf property on these contrasting
soil types and examined the ability of reflectance spectroscopy to quantify leaf chemical and structural traits.
Our specific questions were: (i) what is the relative contribution of soil type and species to leaf trait variation?
(ii) does the importance of the three functional groups (light capture and growth, leaf structure and defence, as
well as rock-derived nutrients and secondary elements) change due to soil or more due to species variation? (iii)
What traits can be accurately and precisely estimated using spectroscopy in temperate woodlands? (iv) To what
extent can leaf spectra be used to detect inter-soil and inter-specific variation in traits?

**2 Material and methods**

**2.1 Field site and sampling**
Leaves were collected from trees growing on deep alluvial soils and shallow chalk soils, near Mickleham in
Surrey (Latitude = 51.26, Longitude = 0.32). The alluvial soil, along the banks of the river Mole, was a loam of
several metres depth. The chalk soil was located on a steep south-facing escarpment into which the river was
cutting; the top soil was a few centimetres deep, underlain by solid chalk (i.e. a typical rendzina soil). The chalk
soils were alkaline with a pH of $7.9 \pm 1.0$ (n = 10), whereas the alluvial was near neutral having a pH of $6.7 \pm$
$0.2$ (n = 10). Phosphorus becomes unavailable to plants in alkaline chalk soil (Gerke, 1992), and much greater
depth of loamy soil on the alluvial surfaces must result in much greater availability of nutrients to plants.
Leaves of 66 trees of six species were collected from the two contrasting soil types. The six species
were in common to both sites: *Acer campestre* L. (Field Maple), *Acer pseudoplatanus* L. (Sycamore), *Corylus*
*avellana* L. (Hazel), *Crataegus monogyna* Jacq. (Hawthorn), *Fraxinus excelsior* L. (Ash) and *Sambucus nigra*
L. (Elder). Two fully sunlit branches were selected, were cut and placed on ice in a cool box, and transported to
a lab for processing within 2 hours (and often within 30 minutes). For each branch, ten mature leaves were
selected. Three samples of 15 leaf disks were cored from these leaves using a 6 mm corer, wrapped in
aluminium foil and frozen in liquid N for later chemical analyses. Leaf areas were measured from fixed-height
photos against a white background analysed in *imageJ*. The scanned leaves were weighed to give hydrated
mass, then dried at 70 °C for a minimum of 72 h to obtain dry mass. Leaf mass per area (LMA) was calculated
as dry mass per unit of fresh leaf area. A further 23 leaf chemical traits were measured on these samples (see
below).

**2.2 Chemical assays**
Protocols for chemical assays are adapted from those developed by the Carnegie Airborne Observatory (see
http://spectranomics.ciw.edu). They are outlined here, with full details available in Supplementary Information.
Oven dried leaves were ground and analysed for a variety of elements and carbon fractions. Concentration of
elements (B, Ca, K, Mg, Mn, P, Si, Fe, Zn) were determined by ashing samples in a muffle furnace, digesting
them in nitric acid, then running them through an inductively-coupled plasma mass spectrometry (Perkin Elmer
SCIEX, Elan DRCII, Shelton, CT, USA). Nitrogen and carbon concentrations were determined using a Thermo
Finnigan 253 with elemental analyser using a gas chromatographic separation column linked to a continuous



flow isotope ratio mass spectrometer. This technique provided foliar concentrations of the stable isotopes of N
and C. Carbon fractions, including hemicellulose, cellulose, lignin and soluble carbon (mainly carbohydrates,
lipids, pectin and soluble proteins), were determined by sequential digestion of increasing acidity (Van Soest,
1994) in an Ankom fiber analyzer (Ankom Technology, Macedon, NY, USA). These carbon fractions are
presented on an ash-free dry mass basis. Concentrations of photosynthetic pigments (chlorophyll *a*, *b*,
anthocyanins and total carotenoids) were measured by spectroscopy of solution derived from frozen leaf disks
on area basis. Absorbance values of the supernatant were measured at wavelengths 470 nm, 649 nm and 665 nm
for chlorophyll *a*, *b* and total carotenoids determination and published equations used to calculate pigment
concentrations as in Lichtenthaler (1987). Absorbance values were also measured at wavelengths 530 nm and
650 nm for anthocyanins determination and published equations used as per Giusti et al. (1999), but corrected
for possible chlorophyll contamination as per Sims and Gamon (2002). The maximum efficiency of
photosystem II (PSII) was calculated according to Genty  et al. (1989) by measuring the maximum fluorescence
(Fm) and the yield of fluorescence in the absence of an actinic (photosynthetic) light (Fo) using a PAM
fluorometer. Total phenolic concentration of the upper methanol/water layer was determined colorimetrically
using the Folin-Ciocalteau method, based on absorbance at 760 nm on a spectrophotometer, and quantified
using tannic acid equivalents with water serving as a blank as per Davey et al. (2007).

### 2.3 Leaf and canopy spectroscopy

The remaining leaves were detached from the branches, and 10 leaves selected at random, avoiding damaged
and soft/young leaves.  These leaves were laid on a matt black surface. Reflectance within bands ranging from
400–2500 nm was measured using a FieldSpec 4, produced by Analytical Spectral Devices (ASD). The
spectrometer's contact probe was mounted on a clamp and firmly pushed down onto the sample, so that no light
escaped through the sides.  The spectral measurements were taken at the mid-point between the main vein and
the leaf edge, approximately half-way between the petiole and leaf tip, with the abaxial surface pointing towards
the probe. The readings were calibrated against a Spectralon white reference every 5 samples. In all statistical
analyses, the mean reflectance values of the 10 measurements per branch were used.

### 2.4 Statistical analyses

Analyses were performed within the R statistics framework (R Core Team 2014). Analyses of variance
(ANOVA) were used to examine the influences of species and soil type on each of the 26 leaf traits. Species,
soil and soil x species terms were included in the model, and the ratio of sum of squares of these terms versus
the total sum of squares was used as an index of species- versus site-level variation. This partition of variance
represent the variation between species, the influence of soil, the interaction between soil and species, and the
unexplained variance referred as to residual variance, which is a combination of intraspecific variation, micro-
site variability, canopy selection and analytical error. Where necessary, variables were log transformed to meet
assumptions of ANOVA.

To evaluate the influence of soil and species on allocation of traits associated with (a) light capture and

growth, (b) defence and structure and (c) rock-derived nutrients and secondary elements, permutational non-
parametric multivariate analysis were performed (Anderson, 2001). This is an analysis of variance using
distance matrices calculated using the *adonis* function in the *vegan* package of R. We recognise that grouping
leaf properties into functional classes can be controversial, given that a single leaf property can contribute to



more than one class (e.g. LMA is related to growth but also to defence). We also performed principal
component analysis (PCA) for each functional class using the function *prcomp* in R.  The principal components
for the variables were obtained by the correlation matrix modelling *in lieu* of covariance matrix modelling, and
then we used the unit variance scaling as in van den Berg et al. (2006) to avoid the effects of variables with high
variance.

Partial least squares regression  (PLSR) was used to evaluate whether field spectroscopy can reliably

estimate leaf properties (Haaland and Thomas, 1988).  There is strong co-llinearity in spectral reflectance data.
PLSR involves dimensionality reduction, producing orthogonal uncorrelated latent vectors containing the
maximum explanatory power in relation to the trait data (Wold et al., 2001). The number of latent variables (nL)
used in the PLSR analysis was estimated by minimising the Prediction Residual Error Sum of Squares (PRESS)
statistic to avoid overfitting (Chen et al. 2004), however was set from 1 to 10 to avoid over-fitting (Zhao et al.,
2015). We adopted a leave-one-out cross-validation for each PLSR model and evaluated the model performance
using coefficient of determination ($R^2$) and root mean square error (RMSE). We also standardised RMSE to the
percentage of the response range (RMSE%) by dividing each RMSE by the maximum and minimum values of
each leaf trait, as in Feilhauer et al., 2010.

The spectral reflectance curve of each sample was transformed into pseudo-absorption (log [1/ R]),

where R is reflectance, based on previous studies (Bolster et al., 1996; Gillon et al., 1999; Richardson and
Reeves, 2005; Petisco et al., 2006; Kleinebecker et al., 2009; Serbin et al., 2014). We reviewed past studies
(Curran, 1989; Elvidge, 1990; Kokaly et al. 2009) to select well documented regions of the spectrum for
absorption features as a basis for predicting each leaf trait. The visible (VIS, 400-700 nm), near infra-red (NIR,
700-1500) and shortwave infra-red I (SWIR I, 1500-1900), shortwave infra-red II (SWIR II, 1900-2500)
regions, as well as combinations of the regions (700-1100 nm, 700-1900 nm, 700-2500 nm, 1100-1500 nm,
1100 -1900 nm, 1100-2500 nm, 1500-2500 nm and 400-2500 nm) were tested and selected based on the model
that minimised RMSE.

**3 Results**

**3.1 Soil and species controls on leaf properties**
Relative foliar concentrations of the macronutrients N, P and K were 17 %, 43 % and 24 % higher on alluvial
compared to chalk soils (Table 1). Nitrogen isotope discrimination ($\delta^{15}$N) varied greatly between the two soils,
from -3.8 ‰ in the chalk soil to 3.4 ‰ in the alluvial. However, foliar concentrations of nutrients required in
smaller quantities (Si, Ca, Mg, B, Mn and Zn) showed the opposite trend: they were higher in chalk soils (by
22%, 37%, 50%, 19%, 23% and 49%, respectively). Fe was the only mineral nutrient unaffected by soil type.
The percentage contribution of soluble C was affected by soil, with an increase in soluble C of 9 % in the
alluvial soil, whereas hemicellulose, cellulose, lignin and LMA were completely unaffected by location.
Carotenoids had 25 % higher concentration in alluvial soil; however other pigments and traits related to water
status ($\delta^{13}$C and water content) varied little with soil type. The efficiency of PSII, which is related to carbon
fixation under controlled conditions, showed a slight increase of 4 % in alluvial soil.

Most traits varied greatly among species and that variation was far greater than the soil effects (Fig. 1).

Interspecific variation ( Green, Fig. 1) accounted for $\geq$ 60% of the variation of eight traits (in descending order
Si, water content, B, soluble C, N, LMA, K and cellulose concentrations), and $\geq$ 40% of the variation of another





six traits (in descending order, lignin, hemicellulose, Mg, Zn, phenolics and Fe). Species exerted little or no
influence on pigment concentrations, efficiency of PSII, $\delta^{13}$C, $\delta^{15}$N, P, Ca and Mn concentrations. The
interaction between species and soil (Blue, Fig. 1) explained little variation and were significant for $\delta^{15}$N, P, Mn
and Zn, but for no other traits. The pigments, efficiency of PSII and $\delta^{13}$C represented the largest unexplained
variance.

**3.2 Variation among functional groups of traits**
Species identity explained 59% of the investment in traits related to defence and structure and 31% of variation
in investment in rock-derived nutrients and secondary elements altogether, but exerted no influence on the
investment in light capture and growth (expressed as $R^2$ values in Table 2). By contrast, soil type explained 6%
of the variation in the rock-derived nutrients with no influences on other functional group. There was an
interaction between soil and species for properties related to the latter group only, which explained 19% of the
total variability in the foliar properties. These results indicate that some species have to invest more in defence
than others regardless of the soil type, whereas soil is an important modifier of traits related to allocation of
macro and micronutrients to the leaves, even though species identity still play an important role in foliar traits
variation for this group.

For leaf properties associated with light capture and growth, the first principal component (PC1)

represents the variation in pigments and investment in light capture, and explains 38% of the total variability,
whereas the second principal component (PC2) represents the variation in water, N, $\delta^{15}$N and soluble C, which
is related to investment in growth, and explains 25% the variability (Fig. 2). The heterogeneity within species
along the PC1 axis tends to be large for all the species, whereas the variation within species along the PC2 tends
to be considerably smaller. Investment in light capture is not species-oriented and also unaffected by soil
variation.

For defence and structure, PC1 represents the lignocellulosic biomass explaining 51% of the total

variability, whereas the PC2 represents LMA and phenolics explaining 21% the variability. Thus, it is possible
to observe a separation of species into two main defensive strategies based on the type of defence. The PC1
distinguish, regardless of the soil type, the species into groups regarding the concentration of lignocellulosic
biomass and Si. The PC2 distinguishes another two groups of species that are also not separated into soil type
regarding the phenolic concentration and LMA.

For macro and micronutrients variation, PC1 represents the mineral nutrients required in greater

amounts explaining 27% of the total variability of leaf properties, whereas the PC2 represents some
micronutrients required in smaller quantities explaining 25% the variability. These 2 axes together explain 52%
of the total variation and can be used to cluster soil into 2 groups: alluvial soils with high P and K concentration
and chalk soil with high B, Mn and Zn concentrations. The inter-specific variation is greater along the PC1
related to Ca, Mg, Fe and B concentrations and can be used to group species.

**3.3 Spectroscopy of leaf properties**
Ability to predict leaf traits from hyperspectral reflectance varied greatly among the 24 traits fitted using the 6
species (Table 3). The number of latent variables ranged from 3 to 9. The $R^2$ obtained varied between 0.16 and
0.92, and RMSE% between 6.9% and 28.7%. PLSR modelling for LMA, water, Si, phenolics, carotenoids, K,
B, efficiency of PSII, N, chlorophyll *a* and chlorophyll *b* were in descending order the best performing in terms



of $R^2$. The highest RMSE % values were for LMA, water, phenolics, carotenoids, K, lignin, B, Si, chlorophyll *a*
and efficiency of PSII. Some minerals, such as P, Zn and Mn, as well as $\delta^{13}C$ and $\delta^{15}N$ showed low $R^2$ and
RMSE% values.

The majority of leaf properties showed higher goodness-of-fit using the regions of the spectrum

between 1100 and 2500 nm. Pigments were the only traits that predictions were more accurate when using the
visible region (400 – 700 nm). Predictions of phenolics, B, Zn and Mn were more accurate with the use of the
region in the SWIR I between 1500 and 1900 nm, whilst Mg needs the use of SWIR II (1900-2500 nm) only.
LMA showed higher $R^2$ and lower RMSE when using the spectrum region between 1100 and 2500 nm, as did
Si, N, soluble C, hemicellulose, cellulose, lignin, $\delta^{13}C$ and $\delta^{15}N$. When using both SWIR regions, higher
goodness-of-fit were obtained for K, Ca and P. Fe was the only foliar property that required the spectrum region
between 700 and 2500 nm.

There were strong correlations among some of the leaf properties (Fig. 3) that can be potentially

leveraging the estimation of other leaf traits from the use of PLSR. The correlation graphic also shows the
similarity among variables through cluster analysis.

**3.4 Use of spectroscopy to distinguish environmental and inter-specific variation**
PLSR models of reflectance data were able to estimate differences in traits among species and detect intra-
specific variation (Fig. 4). In general, inter-specific variation estimated foliar traits quantities reasonably well, as
did for the unexplained variance of most traits. The soil importance was precise for the majority of leaf
properties, but PLSR did not detect precisely the variation of rock-derived nutrients concentration in the leaves
due to soil differences. The use of PLSR also considerably underestimated the importance of soil (~ 37 %) on
the $\delta^{15}N$ variation, but the result was not shown in the graphic (see in *soil*, Fig. 4) due to visual aspects. The
species x soil interaction effects were detected by PLSR modelling, except for traits that showed strong
interaction (Mn, P and $\delta^{13}C$).

**4 Discussion**
Some leaf traits were strongly influenced by both species and soil type, while others were hardly affected by soil
and only varied with species. Soil had a strong influence on concentrations of mineral nutrients in the leaves.
Other foliar properties – mostly those involved in structure, defence and growth - varied among species but soil
had little detectable effect. It is important to emphasize that only fully sunlit leaves were included in the
analyses; as LMA, protein and pigment concentrations are strongly influenced by light environment, sampling
understory leaves would have given a different result.

**4.1 Phenotypic variation associated with soil**
Our findings that trees growing on the chalk soils had relatively low concentrations of N, P and K in their
leaves, and relatively high concentrations of Ca, Mg, B, Mn, Si and Zn, is consistent with previous analyses of
mineral nutrition in calcareous soils. Thin chalk soils contain small quantities of macronutrients needed by
plants, and are unproductive for growing crops unless heavily fertilized; however, cation exchange sites in the
soil contain high concentrations of calcium and magnesium (Hillier et al., 1990). Soil pH has a strong influence
on the plant-availability of many micronutrients: for instance, Zn is readily adsorbed at high pH and forms
organic Zn-ligand complexes at low pH (Broadley et al., 2007). Species that specialize on chalks (so-called





calcicole species) have developed mechanisms for tolerating alkaline soils, associated with low phosphorus
availability and excessive Ca and Mg supply (Misra and Tyler, 2000; Tyler, 2002).
$\delta^{15}N$ discrimination was strongly influenced by soil type, increasing from -3.83 in the chalk soil to 3.43
in the alluvial soil, resulting as the most sensitive foliar trait to soil changes. Although the species *Alnus*
*glutinosa* (L.) Gaertn. was not included in the field measurements for trait determination, this species was
restricted to alluvial soils in our study area and may help explain some differences in leaf traits between soils.
The species *Alnus glutinosa* is an N fixing plant and is known to be dependent on mycorrhizal fungi (Hall et al.,
1979) and the most important benefit of mycorrhizae is an increase in the efficiency of nutrient uptake by plants,
especially phosphorus. Variation in $\delta^{15}N$ among plants within an ecosystem has been interpreted as representing
differences in fixation, mycorrhizal dependence, depth of acquisition within the soil profile, utilization of
depositional N and the form of N that plants predominantly acquire (Vallano and Sparks, 2013).
The discovery that structural and defensive traits do not vary with soil is consistent with a previous
study in New Zealand's lowland temperate rain forests (Wright et al., 2010). That study compared traits of trees
growing on phosphorus rich alluvium versus phosphorus-depleted marine terraces. Foliar phosphorus
concentrations of species were halved on the marine terraces, but there was no detectable variation in structural
traits, phenolic or tannin concentrations.

**4.2 Inter-specific and residual variation**
Species had a greater influence on trait values than soils for all traits, except P. The traits most influenced by
species (in descending order) were Si, water, B, soluble C, N, LMA, K, cellulose, lignin, hemicellulose,
magnesium, Zn, phenolics and Fe. It is interesting to note that two trace elements were near the top of this list; it
is likely that strong differences in B and Si concentrations among species reflect differences in ion channel
activity in roots (Ma and Yamaji, 2006). Previous studies have also shown Si to be under strong phylogenetic
control, and to be little affected by environmental conditions (Hodson et al., 2005). We also found Si and B
concentrations to be positively correlated, which might ameliorate the effects on B toxicity as Si can increases B
tolerance of plants (Gunes et al., 2007). High Zn organization at the species level corroborates earlier analysis
that show more than 70% of Zn variation occurs between and within species (Broadley et al., 2007). Structural
foliar traits and more expensive compounds were also found to have high interspecific variation, such as
cellulose and lignin, suggesting that even on a strong soil filtering, species play the crucial role to invest in these
specific traits.
The residual variation is a combination of intraspecific variation, micro-site variability, canopy
selection and measurement error. The residual variation was high for $\delta^{13}C$ and pigments, greatly exceeding soil
and species effects, as also reported for pantropical trait studies (Asner and Martin, 2011). Low coefficient of
variation in $\delta^{13}C$ among samples, and high residual variation, suggest that the efficiency of C fixation is
maintained among species and soil. $\delta^{13}C$ is known to vary strongly with light condition and relative humidity
(Yan et al., 2012), but their study sampled only from fully sunlit leaves.

**4.3 Functional groups on contrasting soils**
We investigated how traits in generalist species are responding to different soil conditions and the factors most
contributing to changing leaf properties. The investment in light capture had high intra-specific variation, and
neither species nor soil accounted for variation in foliar properties. The investment in growth showed relative



high inter-specific variation separating out some species. Investment in traits related to defence and leaf
structure is species-mediated, and may be separated into two defensive strategies. Considering these traits, some
species invest more in LMA and phenolics and other species invest more in lignocellulosic biomass and Si
regardless of soil type. The allocation of rock-derived nutrients to leaves is highly dependent on soil as
environmental filter.
Traits favouring high photosynthetic rate and growth are considered to be advantageous in rich-
resource soil environments, whereas expressions of traits favouring resource conservation are considered
advantageous in low-resource environments (Aerts and Chapin, 1999, Westoby et al., 2002). Nevertheless, Fine
et al. (2006) found similar results to ours with seedlings transplantation for 6 species into different soil types,
concluding that investment in defence is due to genetically based, fixed traits, and defence differences are not
just passive responses to differences of available nutrients in the soils.

**4.4 Predictions of foliar traits using spectroscopy**
Several leaf chemical traits and LMA could be estimated accurately using visible-to-shortwave infrared
spectroscopy. Previous studies have also shown that leaf spectra can be used to predict leaf chemical properties
(Asner and Martin, 2008, Asner and Martin, 2009; Asner et al., 2015). Doing so revealed that LMA, water, Si,
total phenolics, carotenoids and K produced the most consistent and accurate calibrations.
The locations of important wavelengths in our PLSR models match the locations of known spectral
absorption features related to proteins, starch, lignin, cellulose, hemicellulose and leaf water content (Kokaly et
al., 2009). In the region between 700 and 2500 of the electromagnetic spectrum, absorption features are
commonly the result of overtones and combinations of fundamental absorptions at longer wavelengths. The
visible region was useful to predict pigments concentrations and the efficiency of PSII only, whereas the infra-
red region was associated with most traits. The region of importance with correlated wavelengths with nitrogen
varies between 1192 nm in deciduous forest (Bolster et al., 1996) to 2490 for forage matter (Marten et al.,
1983), which results directly from nitrogen in the molecular structure. Although chlorophylls also contain
nitrogen, the spectra of chlorophylls differ greatly from proteins because of their dissimilar chemical structures,
showing strong absorption due to C-H bonds in the phytol tail of the molecule (Katz et al., 1966), also
confirmed in this work when visualizing the regions of importance for predictions. The 1500-1900 nm region
was also important for phenolic compounds prediction, which includes the 1660 nm feature across a variety of
species and phenolic compounds (Windham et al., 1988; Kokaly and Skidmore, 2015).
A review in the literature suggests that the use of dry leaves may improve predictions of lignocellulosic
biomass in the leaves with the use of spectroscopy (Richardson and Reeves, 2005; Asner et al., 2011; Serbin et
al., 2014), as the strong water absorption features mask most of the biochemical absorption features (Fourty and
Baret, 1998). On the other hand, the use of spectroscopy on fresh leaves is particularly better for LMA
predictions, given the strong coupling between water content, leaf structure and LMA (Asner et al., 2011). The
primary and secondary effects of water content on leaf reflectance are greatest in spectral bands centred at 1450,
1940, and 2500 nm (Carter, 1991), but has also been predicted using bands between 1100-1230 nm absorption
features (Ustin et al., 1998; Asner et al., 2004).
The use of spectroscopy for Si predictions on fresh leaves appears to be promising considering our
accurate results. The data available in the literature show that the ecological functions of Si have generally been
poorly studied, and that there are almost no data about the role of Si structures in the reflection and transmission





spectra of short-wave or photosynthetically active radiation in plants. Silicon is absorbed by plants from the soil
solution in the form of silicic acid (H4SiO4) being translocated to the aerial parts of the vegetal through xylema,
and then deposited along the plant as phytoliths (silicified bodies) (Tripathi, 2011). Smis et al. (2014) showed
for the first time the potential use of NIR spectroscopy to predict Si concentration. Si shows strong interactions
with plant biomolecules such as phenol- or lignin-carbohydrate complexes (Inanaga et al., 1995), cellulose (Law
and Exley, 2011), and proteins (Perry and Keeling-Tucker, 2003). Predictions of Si concentrations, and other
traits, from leaf spectra reflectance can be stronger than expected likely because leaf spectra integrate
information on several foliar traits simultaneously.
Galvez-Sola et al. (2015) showed that near-infrared spectroscopy can constitute a feasible technique to
quantify several macro and micronutrients such as N, K, Ca, Mg, Fe and Zn in citrus leaves of different leaves
with coefficient of determination ($R^2$) varying between 0.53 for Mn and 0.99 for N, whereas B showed less
accurate results with the use of spectroscopy. The regions of importance for prediction were relatively similar to
all the mineral nutrients analysed in this study, except for B that had the band between 1500 and 1900 as the
best predictive region. Similar to Si, the relative high precisions for K, Fe and B predictions can be stronger due
to the integrating information on several foliar traits simultaneously.

### 4.5 Consideration on the use of spectroscopy to quantify patterns of foliar traits

The range of variation within species for most predicted traits tend to be smaller with the use of PLSR on
reflectance, resulting in consistent slight overpredictions of the inter-specific variance. The interrelationships
between foliar chemical and spectral properties for each species help to explain the successful results reported in
developing species-level variation from leaf spectral data (Asner et al., 2009). In general, the residuals variation
was lower for most leaf traits with the use of spectroscopy, possibly because the use of spectroscopy affects the
ability to quantify measurement error, one of the residual variation components.
The variation caused by soil on mineral nutrients and $\delta^{15}N$ allocated to the leaves remained unchanged
with the use of spectroscopy, possibly because structural leaf traits, such as LMA, cellulose, water, as well as
pigments, contribute more to leaf reflectance. As these structural traits remained unchanged between soil types
for the six species, it possibly explains why the analyses were not able to detect the mineral nutrients and $\delta^{15}N$
effects on reflectance, considering that spectroscopy sensitivity to these properties are an artefact of traits
correlation rather than a real feature. The same occurs when accounting for variation related to the interaction
between soil and species. The soil component in the interaction tends to be underestimated for rock-derived and
$\delta^{13}C$.
This study particularly provides findings for a large range of traits that indicate that the use of
spectroscopy may be useful to quantify structural traits but can be misleading to measure the environmental
filtering on traits that are indirectly predicted, such as macro- and micronutrients. While remote sensing is not a
direct replacement of field sampling, the ability of remote sensing platforms to assess biological phenomena at
large spatial scales is unparalleled.

### 5 Conclusions

Analyses of trait variation shows that the identity of the species has a much stronger influence on most traits
than the substrate upon which the tree grows. Traits associated with light capture, cell wall structure and defence
were particularly uninfluenced by substrate, while rock-derived nutrients are strongly influenced by the soil



characteristics. This study also demonstrates the potential for estimating foliar traits by field spectroscopy and
its promising use to predict Si. LMA, water, N, pigments, phenolics, K, B and hemicellulose were also
accurately estimated at the species level. However, subtle changes in traits associated with soil type were not
generally detectable, possibly because the spectroscopy sensitivity to these traits is an artefact of correlation
with other traits that did not change due to soil type.

**Authors' Contributions**
MHN participated in the chemical analysis, analysed the data and wrote the manuscript; MPD led the chemical
analysis and wrote the manuscript; DAC conceived the ideas, designed methodology, collected the data and led
the writing of the manuscript. All authors contributed critically to the drafts and gave final approval for
publication

**Acknowledgments**
We thank undergraduate students Thomas Hitchcock, Lilian Halstead, Matt Chadwick and Connor
Willmington-Holmes for helping with field work, and Alexandra Jamieson for measuring phenolics. We are
grateful to David Burslem for providing access to carbon fractions analyser at the University of Aberdeen. The
field spectrometer was hired from the NERC Field Spectroscopy Facility; we are grateful to Alasdair MacArthur
for his training and advice. Matheus H. Nunes is supported by a PhD scholarship from the *Conselho Nacional*
*de Pesquisa e Desenvolvimento* (CNPq).

**Competing interests**
The authors declare that they have no conflict of interest.

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





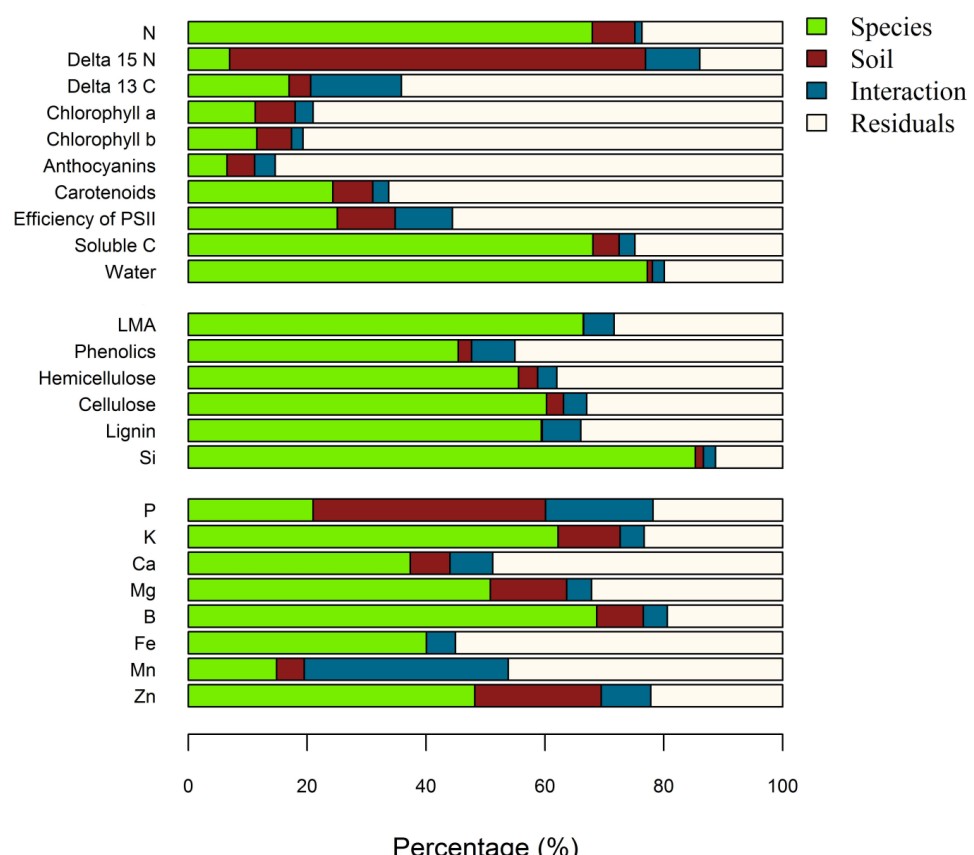

**Figure 1.** Partitioning of variance of foliar properties between species, soil, species x soil interaction and residual components for six generalist species found on both chalk and alluvial soils. Residual variation arises from within-site intraspecific variation, micro-site variability, canopy selection and measurement error variance.



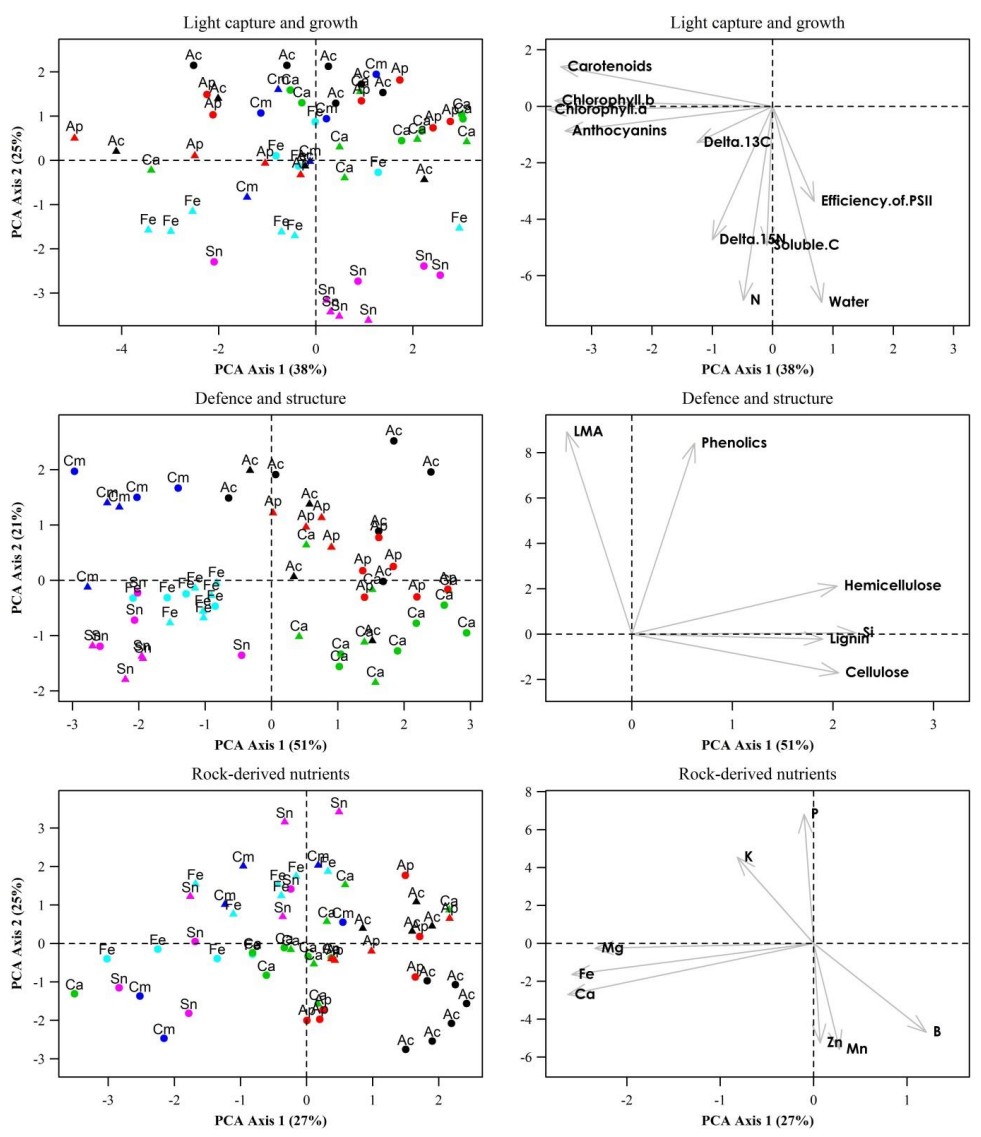


**Figure2**. Principal component analysis of traits related to light capture and leaf hydraulic, defence and leaf
structure, and metabolism and maintenance. Fe = *Fraxinus excelsior*; Sn = *Sambucus nigra*; Ac = *Acer*
*campestre*; Cm = *Crataegus monogyna*; Ca = *Corylus avellana*; Ap = *Acer pseudoplatanus*; Δ = alluvial soils;
and ○ = chalk soils.



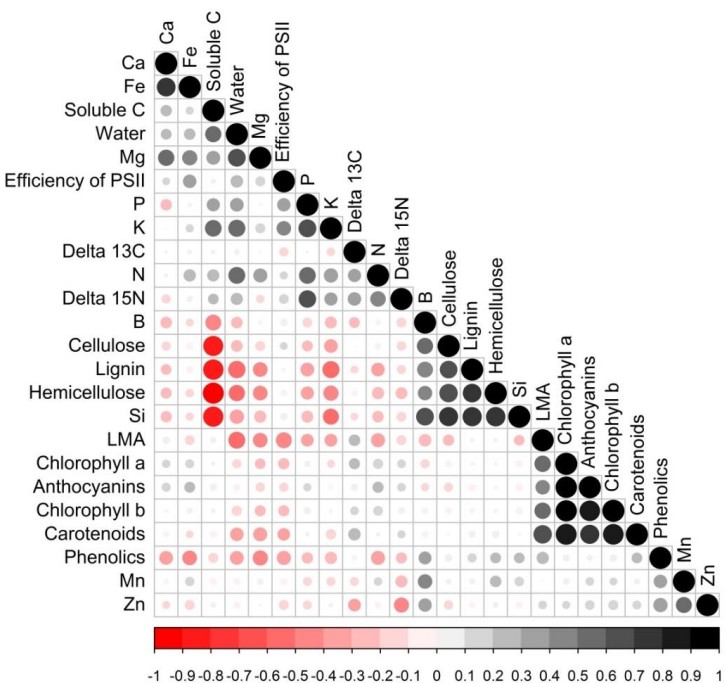


**Figure 3.** Spearman correlation rank test among leaf traits of 6 species growing on both soil types. Red and

black circles mean negative and positive correlations. Foliar traits were organised using cluster analysis.






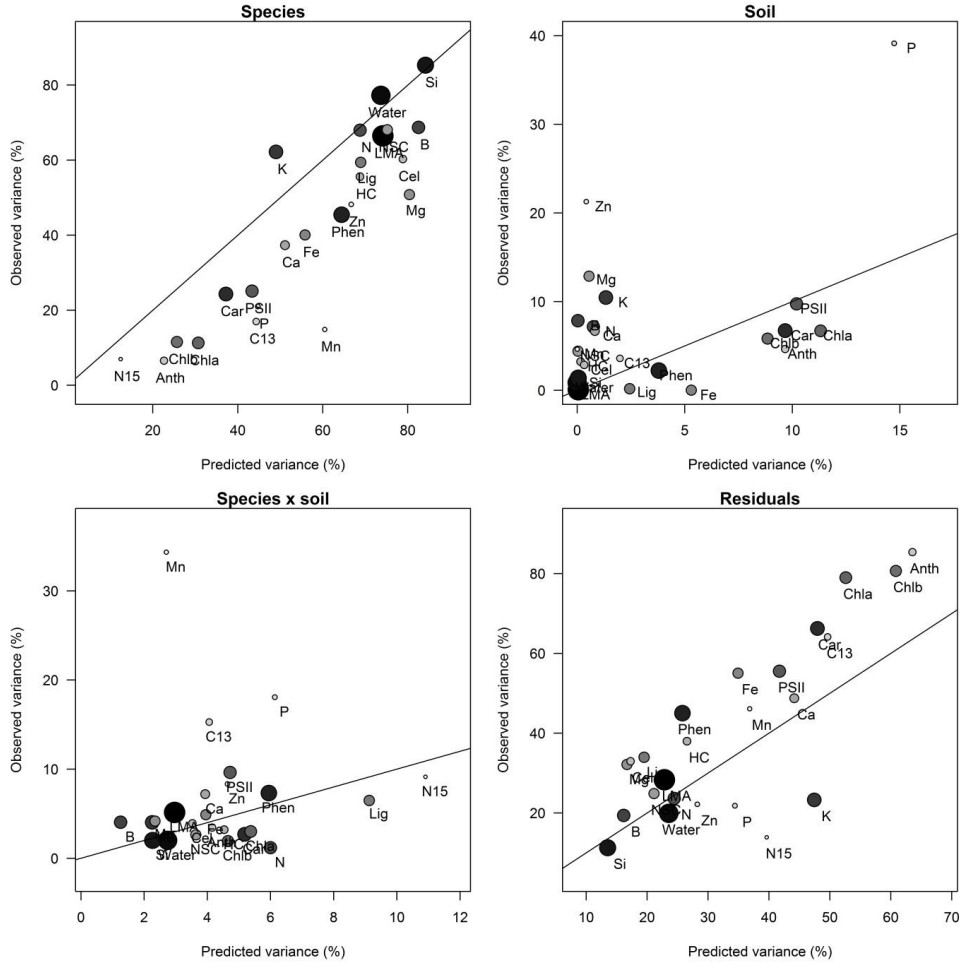


**Figure 4.** Predicted values from PLSR on reflectance versus actual partitioning of variance in foliar properties

between species, soil, species x soil interaction and residual (intraspecific variation, micro-site variability,

canopy selection and measurement error) variance, for six generalist species found on both chalk and alluvial

soils. The greyness and size of each dot reflects the goodness-of-fit of the PLSR for each foliar trait, with darker

and bigger points representing the most accurate PLSR predictions.








**Table 1.** Average, standard deviation (SD) and coefficient of variation (CV) in percentage for leaf traits of six
generalist species growing on alluvial and chalk soils. Foliar property was statistically different between soil
types with $P$-value < 0.05 *, < 0.01 ** and < 0.001 ***.

| Properties | Alluvial | | Chalk | |
|---|---|---|---|---|
| | Mean ± SD | CV | Mean ± SD | CV |
| **Light capture and growth** | | | | |
| N (%) *** | 2.53 ± 0.81 | 32.1 | 2.16 ± 0.73 | 34.0 |
| $\delta^{15}$N (‰) *** | 3.43 ± 2.65 | 77.3 | -3.83 ± 2.01 | 52.3 |
| $\delta^{13}$C (‰) | -28.2 ± 1.2 | 4.5 | -28.7± 1.0 | 3.6 |
| +Chlorophyll a (mg m$^{-2}$) | 338.8 ± 116.0 | 34.2 | 279.6 ± 89.2 | 31.9 |
| Chlorophyll b (mg m$^{-2}$) | 78.6 ± 27.6 | 35.1 | 64.7 ± 22.4 | 34.7 |
| Anthocyanins (mg m$^{-2}$) | 423.3 ± 143.8 | 33.9 | 362.8 ± 121.6 | 33.5 |
| Carotenoids (mg m$^{-2}$) * | 110.5 ± 40.4 | 36.5 | 88.2± 35.5 | 40.2 |
| Efficiency of PSII ** | 0.74 ± 0.05 | 7.1 | 0.71 ± 0.06 | 9.8 |
| Soluble C (%) ** | 73.6 ± 6.5 | 8.8 | 70.3 ± 7.5 | 10.6 |
| Water (%) | 59.1 ± 8.2 | 14.0 | 58.5 ± 7.9 | 13.5 |
| **Defence and structure** | | | | |
| +LMA (g cm$^{-2}$) | 60.8 ± 24.0 | 39.4 | 60.6 ± 23.6 | 38.9 |
| Phenolics (%) | 83.7 ± 64.1 | 76.5 | 84.3 ± 49.7 | 59.0 |
| +Hemicellulose (%) | 10.9 ± 3.2 | 29.8 | 12.5 ± 3.6 | 29.4 |
| Cellulose (%) | 10.1 ± 1.8 | 18.6 | 11.0 ± 2.1 | 19.3 |
| Lignin (%) | 3.9 ± 1.9 | 49.8 | 4.7 ± 3.1 | 64.8 |
| +Si (%) * | 0.91 ± 0.56 | 62.2 | 1.11 ± 0.79 | 71.5 |
| **Rock-derived nutrients** | | | | |
| +P (%) *** | 0.20 ± 0.05 | 25.5 | 0.14 ± 0.03 | 26.8 |
| K (%) *** | 0.98 ± 0.49 | 50.0 | 0.79 ± 0.50 | 64.4 |
| +Ca (%) * | 1.67 ± 0.75 | 45.1 | 2.29 ± 1.24 | 54.1 |
| +Mg (%) *** | 0.24 ± 0.11 | 47.1 | 0.36 ± 0.15 | 43.8 |
| +B (µg g$^{-1}$) *** | 29.0 ± 8.7 | 30.1 | 34.5 ± 12.4 | 36.0 |
| +Fe (µg g$^{-1}$) | 122.3 ± 24.6 | 20.1 | 125.4 ± 32.0 | 25.5 |
| +Mn (µg g$^{-1}$) * | 84.7 ± 64.3 | 75.9 | 103.8 ± 69.5 | 66.9 |
| +Zn (µg g$^{-1}$) *** | 22.9 ± 12.6 | 55.0 | 34.1 ± 18.7 | 54.9 |

+log transformed prior to ANOVA.








**Table2**. Permutational multivariate analysis to calculate the partitioning of variance in set of foliar traits related
to each functional class between species, soil, species x soil interaction and residual variance for six generalist
species found on both chalk and alluvial soils. All differences were significant ($P$-value $< 0.05$ *, $< 0.01$ ** and
$< 0.001$ ***) unless indicated as not significant (NS).

| Component | Light capture and growth | | Defence and structure | | Rock-derived nutrients | |
|---|---|---|---|---|---|---|
| | $F$-test | $R^2$ | $F$-test | $R^2$ | $F$-test | $R^2$ |
| Species | $1.48^{ns}$ | 0.13 | 14.9*** | 0.59 | 6.1*** | 0.31 |
| Site | $2.96^{ns}$ | 0.05 | $0.84^{ns}$ | 0.00 | 5.6** | 0.06 |
| Interaction | $0.43^{ns}$ | 0.04 | $1.23^{ns}$ | 0.05 | 3.8*** | 0.19 |
| Residuals | | 0.78 | | 0.34 | | 0.41 |

































**Table3.** Partial Least Squares Regression (PLSR) on spectral data and leave-one-out cross-validation for 24 leaf
traits of 6 species occurring on both alluvial and chalk soils. The model calibration and validation performance
was evaluated for each leaf property by calculating the coefficient of determination ($R^2$), root mean square error
(RMSE) and the percentage root mean square error (%) based on the given number of latent variables (nL) for
each PLS model.

| Leaf property | Spectrum range (nm) | nL | R2 | | RMSE | | RMSE% | |
|---|---|---|---|---|---|---|---|---|
| | | | Cal | Val | Cal | Val | Cal | Val |
| **Light capture and growth** | | | | | | | | |
| N (%) | 1100 – 2500 | 3 | 0.61 | 0.55 | 0.49 | 0.52 | 15.0 | 16.0 |
| $\delta^{15}$N (‰) | 1100 – 2500 | 9 | 0.41 | 0.16 | 3.28 | 4.01 | 23.5 | 28.7 |
| $\delta^{13}$C (‰) | 1100- 2500 | 6 | 0.46 | 0.30 | 0.85 | 0.96 | 16.1 | 18.2 |
| [+]Chlorophyll *a* (mg m$^{-2}$) | 400-700 | 7 | 0.65 | 0.53 | 60.05 | 69.62 | 13.5 | 15.7 |
| Chlorophyll *b* (mg m$^{-2}$) | 400-700 | 4 | 0.59 | 0.50 | 16.48 | 18.57 | 15.2 | 17.1 |
| Anthocyanins (mg m$^{-2}$) | 400-700 | 4 | 0.45 | 0.33 | 99.20 | 110.70 | 18.0 | 20.1 |
| Carotenoids (mg m$^{-2}$) | 400-700 | 7 | 0.75 | 0.62 | 19.31 | 23.54 | 11.0 | 13.4 |
| Efficiency of PSII | 400-2500 | 6 | 0.68 | 0.55 | 0.03 | 0.04 | 13.4 | 15.9 |
| Soluble C (%) | 1100 – 2500 | 4 | 0.54 | 0.46 | 4.76 | 5.15 | 18.1 | 19.6 |
| Water (%) | 1100 – 1500 | 5 | 0.87 | 0.83 | 2.89 | 3.29 | 9.0 | 10.1 |
| **Defence and structure** | | | | | | | | |
| [+]LMA (g cm$^{-2}$) | 1100 – 2500 | 6 | 0.94 | 0.92 | 1.09 | 1.12 | 6.1 | 6.9 |
| Phenolics (%) | 1500 – 1900 | 6 | 0.78 | 0.70 | 26.20 | 30.48 | 9.7 | 11.3 |
| [+]Hemicellulose (%) | 1100-2500 | 4 | 0.44 | 0.35 | 1.28 | 1.30 | 18.4 | 19.8 |
| Cellulose (%) | 1100-2500 | 4 | 0.44 | 0.34 | 1.52 | 1.66 | 17.0 | 18.6 |
| Lignin (%) | 1100-2500 | 4 | 0.57 | 0.47 | 1.72 | 1.89 | 13.0 | 14.2 |
| [+]Si (%) | 1100 – 2500 | 4 | 0.77 | 0.72 | 1.50 | 1.55 | 14.4 | 15.5 |
| **Rock-derived nutrients** | | | | | | | | |
| [+]P (%) | 1500-2500 | 7 | 0.43 | 0.22 | 1.26 | 1.30 | 17.8 | 20.2 |
| K (%) | 1500 – 2500 | 7 | 0.70 | 0.61 | 0.27 | 0.31 | 11.9 | 13.6 |
| [+]Ca (%) | 1500-2500 | 7 | 0.53 | 0.40 | 1.40 | 1.47 | 15.9 | 17.9 |
| [+]Mg (%) | 1900 – 2500 | 3 | 0.54 | 0.46 | 1.39 | 1.42 | 15.2 | 16.5 |
| [+]B (µg g$^{-1}$) | 1500-1900 | 6 | 0.66 | 0.56 | 1.24 | 1.28 | 13.6 | 15.2 |
| [+]Fe (µg g$^{-1}$) | 700 – 2500 | 5 | 0.56 | 0.46 | 1.17 | 1.19 | 15.6 | 17.2 |
| [+]Mn (µg g$^{-1}$) | 1500-1900 | 6 | 0.35 | 0.20 | 1.83 | 1.95 | 20.5 | 22.7 |
| [+]Zn (µg g$^{-1}$) | 1500-1900 | 7 | 0.41 | 0.21 | 1.50 | 1.60 | 19.5 | 22.4 |

[+] Trait values were natural log-transformed for PLSR.