# Peer review of "On the challenges of using field spectroscopy to measure the"

_Biogeosciences, 2016_

## Referee Comment (RC1) · Anonymous Referee #1 · 7 Dec 2016

General Comments

This is a well written, interesting paper that attempts to use leaf spectroscopy to predict leaf traits in two contrasting soil types. They found that traits tended to be specific to species and that soil type had much less of an influence. They used the PLSR methodology to predict traits with spectroscopy and found reasonably good relationships which reflect previous studies. Overall, this is a solid analysis and asks a relevant question of interest to the readers of this journal. Below I suggest a few areas where the paper could be strengthened and a number of minor points.

Specific comments 1) The Material and Methods 'Statistical Analyses' section needs to be much expanded and clarified. Especially in regards to Figures 3 and 4. Without

knowing how the data for those sections were acquired, it is difficult to evaluate the claims made in the results and discussion section. 2) Some of the findings discussed in the abstract need to be made clearer. 3) Some of the claims/statements made in the abstract and intro either need to be changed or better supported with relevant literature. 4) Many frequently used terms throughout the paper need to be changed/clarified (see below). 5) Why there is such an emphasis on being able to predict Si using PLSR throughout the paper needs to be clarified! 6) Discuss more clearly the relevance of the findings in terms of future high resolution aircraft campaigns. Based on these results, what sort of aircraft data could be produced for temperate forests.

Technical Corrections Terminology

Change uses of "among species' to "between species" (if that is what is meant).

Change uses "species x soil interaction" to "species-soil interaction" or something similar.

Change uses of "goodness-of-fit" to "strength of relationships" or something similar.

Change uses of "leaf property" to "leaf trait".

Abstract/summary

Line 10 – change "traits variation" to "trait variation"

Line 12 – "Hyperspectroscopy is a recently developed technology for estimating the traits of fresh leaves" – disagree (the technique dates back to the 90s – e.g. Curran, 1989)

Line 13 – "Few studies have evaluated its potential for assessing inter- and intra-specific trait variability in community ecology" – This is a contentious claim given a lot of Asner's work (e.g. Asner and Martin, 2011). This statement is not supported in the introduction.

Line 14 – "Working with 24 leaf traits". Contradicted by line 151 which mentions 26 leaf

traits.

Line 19 - "(iv) Can leaf spectra be used to detect inter-soil as well as inter-specific variation in traits?" – I don't understand how this question differs from iii ("what traits can be estimated precisely using field spectroscopy?"). If you can precisely estimate a trait using field spectroscopy, then surely it will be able to detect inter-soil and inter-specific variations. Unless the estimation only works on one species type on one particular soil type. Maybe rephrase?

Line 20 – "The contribution of species and soil-type effects to variation in traits were evaluated using statistical analyses" – maybe state a few of the main statistical analyses used?

Line 21 – "Foliar traits were predicted from spectral reflectance using partial least square regression, and so inter- and intra-specific variation." – Presumed typo – rewrite.

Line 22 – "Most leaf traits varied greatly among species" – a) replace 'among species' with either within or between species (presumed between?) b) Also this sentence is confusing – suggests that there was simply a wide variation in leaf trait measurements - slightly random to mention in abstract. Actual meaning I think is something along these lines "Inter-specific variation was the largest contributor to trait variation".

Line 23 – "Macronutrient concentrations were greater on alluvial than chalk soils while micronutrient concentration showed the opposite trend" – Foliar macronutrient concentrations or soil macronutrient concentrations? (presumed the former?). Also, slightly odd sentence – what's the significance? Maybe meant to say something along these lines? - "However, foliar macro- and micronutrient concentrations were found to be more strongly influenced by soil type".

Line 24 – "Si predictions using spectroscopy appear to be promising" – what's so special about Si predictions?! Why do they get singled out?

Line 28 – "However, it [field spectroscopy] was unable to detect subtle within-species variation of traits associated with soil type" – repetition of line 25? ("Field spectroscopy….was less effective at detecting subtle variation of rock-derived nutrients between soil types"). Combine sentences to keep abstract concise?

Introduction

Line 58 – typo. Change "include phosphorous" to "including phosphorus".

Line 64 – "along environmental change". Typo. Suggested "along environmental gradients"?

Line 71 – "However, spectral and chemical properties may be uncoupled if intraspecific variation in foliar traits is high and/or phenotypic plasticity exceeds phylogenetic patterns among leaf properties". Disagree. Spectral and chemical relationships would still hold, it would just be harder to identify species type based on their reflectance signatures.

Line 73 – "Martin and Aber (1996) demonstrated that equations for estimating leaf properties from one site were unable to predict leaf properties for other sites, due to variability in the magnitudes of foliar traits levels between data sets and environmental influences". Very old reference and what about all the evidence to the contrary (e.g. all of Asner's work) ???

Line 75 – "To our knowledge, the link between foliar traits and spectral properties of trees has not been broadly demonstrated for temperate forests" – query this statement. The remote sensing of foliar traits began in temperate forests.

Line 84 – "leaf property". Replace with "leaf trait"?

Line 86 – "what is the relative contribution of soil type and species to leaf trait variation?". Missed word? "what is the relative contribution of soil type and species type to leaf trait variation".

Line 88 – "does the importance of the three functional groups change due to soil or more due to species variation?" – awkward phrasing. Rephrase.

Material and Methods

Line 102: "Leaves of 66 trees of six species were collected from the two contrasting soil types. The six species were in common to both sites". Suggested "Across both sites, leaves were collected from 66 trees, representing six species. The six species common to both sites were:"

Line 103: "Acer campestre L. (Field Maple)" – what does the L stand for?

Line 105: "Two fully sunlit branches were selected, were cut and placed on ice in a cool box, and transported to a lab for processing within 2 hours (and often within 30 minutes)".

Line 108: "Leaf areas were measured". Suggested "Leaf area was measured?"

Line 149: "2.4 Statistical analyses". Needs to be split up into each statistical analysis performed and titled accordingly.

Line 156: "Where necessary, variables were log transformed to meet assumptions of ANOVA". Reference Table 1, where info concerning which variables were log transformed can be found.

Line 169: "strong co-llinearity". Typo.

Line 168: PLSR section – no mention of using 70% to calibrate and 30% to test but Cal and Val appear on Table 3.

No mention of how the data for Figure 3 and 4 is acquired!!!

Results

Line 204 – "Species exerted little or no influence on pigment concentrations" – Refer to species in this context (and throughout paper) as 'species type"?

Line 241: "Ability to predict leaf traits from hyperspectral reflectance varied greatly among the 24 traits fitted using the 6 species (Table 3)". "fitted using the 6 species" - confusing. Rephrase.

Line 243: "PLSR modelling for LMA, water, Si, phenolics, carotenoids, K, 243 B, efficiency of PSII, N, chlorophyll a and chlorophyll b were in descending order the best performing in terms of"

Line 248- "higher goodness-of-fit" – use a different term? E.g. stronger relationships/correlations etc.

Line 256: "There were strong correlations among some of the leaf properties (Fig. 3) that can be potentially leveraging the estimation of other leaf traits from the use of PLSR". Interesting. Explain further?

Line 257: "The correlation graphic also shows the similarity among variables through cluster analysis". Explain. Cluster analysis was not been mentioned in the Materials and Methods. Explain how this was achieved, why it was done and expand on results.

Discussion

Line 271: "Some leaf traits were strongly influenced by both species and soil type, while others were hardly affected by soil and only varied with species". Vague. Make more specific.

Line 305: "water" – change to 'leaf water content'.

Line 321: "but their study sampled only from fully sunlit leaves". Suggested - "Similarly, their study sampled only from fully sunlit leaves".

Line 325: "The investment in light capture had high intra-specific variation, and neither species nor soil accounted for variation in [these] foliar properties". Missing word.

Line 326: "relative". Typo (relatively)

Line 327: "separating out some species". Confusing. Rephrase?

Line 327: "Investment in traits related to defence and leaf structure is species-mediated, and may be separated into two defensive strategies". State the two defensive strategies?

Line 342: "Doing so revealed that. . .". Awkward. Rephrase.

Line 351: "Although chlorophylls also contain nitrogen, the spectra of chlorophylls differ greatly from proteins because of their dissimilar chemical structures, showing strong absorption due to C-H bonds in the phytol tail of the molecule (Katz et al., 1966), also confirmed in this work when visualizing the regions of importance for predictions." Require a full stop after (Katz et al. 1996) and develop last sentence ("also confirmed in this work when visualizing the regions of importance for predictions").

Line 357: "A review in the literature". "A review of the literature"

Line 360: "On the other hand, the use of spectroscopy on fresh leaves is particularly better for LMA predictions"

Line 365: "The use of spectroscopy for Si predictions on fresh leaves appears to be promising considering our accurate results". Maybe, but why are Si predictions so important? What ecological function does Si perform?!

Line 339: 4.4 Predictions of foliar traits using spectroscopy – this section maybe a bit long? Could condense? Says some interesting things but I'm not sure they're all relevant to the paper.

Line 384: Consideration on the use of spectroscopy to quantify patterns of foliar traits. Typo - Consideration of the use of spectroscopy to quantify patterns of foliar traits.

Line 385. "The range of variation within species for most predicted traits tend to be smaller with the use of PLSR on reflectance". Very confusing. Rephrase.

Line 399: "This study particularly provides findings for a large range of traits that indicate that the use of spectroscopy may be useful to quantify structural traits but can be misleading to measure the environmental filtering on traits that are indirectly predicted, such as macro- and micronutrients". I might agree if I understood Figure 4 but, as I don't, I query this statement.

Line 401: "While remote sensing is not a direct replacement of field sampling, the ability of remote sensing platforms to assess biological phenomena at large spatial scales is unparalleled". Slightly random – doesn't follow from previous statement/results section.

Conclusion

Line 407: "rock-derived nutrients are strongly influenced by the soil characteristics". Need to tone down or change previous sentence, otherwise statements are contradictory.

Line 409: "This study also demonstrates the potential for estimating foliar traits by field spectroscopy and its promising use to predict Si". a) "demonstrates the potential" –this has already been done many times. Maybe something more along the lines of "agrees with the existing literature in demonstrating the potential..." b) "its promising use to predict Si". Once again – what is so important about Si?!?!?!

Figures

Line 661: "Red and black circles mean negative and positive correlations". Which way round?

Line 668: "The greyness and size of each dot reflects the goodness-of-fit of the PLSR for each foliar trait, with darker and bigger points representing the most accurate PLSR predictions. goodness-of-fit". Give statistical boundaries for how dots were sorted into each size/shape category.

Line 675: Table 1. CV needs to be represented as %CV, as stated in the heading.

---

## Referee Comment (RC2) · Anonymous Referee #2 · 16 Dec 2016

The article "Leaf trait variation and field spectroscopy of generalist tree species on contrasting soil types" by Nunes and co-authors analyzed field spectroscopy data collected on different European tree species on contrasting soil types. The authors worked with 24 leaf traits and explored the following questions: What contribution do soil type and species identity make to trait variation? When traits are clustered into three functional groups (light capture and growth, leaf structure and defence, as well as rock-derived nutrients), are some groups more affected by soil than others? What traits can be estimated precisely using field spectroscopy? Can leaf spectra be used to detect inter-soil as well as inter-specific variation in traits? The authors found that most leaf traits varied greatly among species. The effects of soil type were generally weak by comparison

Specific Comments:

Line 28 variation in foliar traits and Si predictions using spectroscopy appear to be promising. Not clear what Si means at this stage, it becomes clear later. But in general all the discussion on Si is poor

Line 162 We recognize that grouping leaf properties into functional classes can be controversial, given that a single leaf property can contribute to This is particularly true for P, this assumption has to be justified as foliar P can be easily considered a trait associated to growth

Results Section Spectroscopy of leaf properties The results of PLSR are on one hand encouraging because the portion of spectra selected for specific traits are in line with what expected from the literature. Some examples from the article:

1)higher goodness-of-fit were obtained for K, Ca and P in the SWIR regions. 2) Pigments were the only traits that predictions were more accurate when using the visible region (400 – 700 nm)

I think would be useful to have more discussion on what is known and what is new compared for instance to the review from Homolova et al., which discuss many of the traits mentioned by the authors and how these traits can be predicted from remote sensing data. What do we learn from these results? I think the authors should make an effort to improve this aspect because can be quite relevant considering the great dataset they have. For example a figure with a reflectance spectra with an indication of the regions relevant to estimate other the traits indicated migh be useful for the reader.

Line 267 The species x soil interaction effects were detected by PLSR modelling, except for traits that showed strong interaction (Mn, P and $\delta$ 13C)

This should be better discussed

Line 279 Our findings that trees growing on the chalk soils had relatively low concentrations of N, P and K in their leaves, and relatively high concentrations of Ca, Mg, B,

Mn, Si and Zn, is consistent with previous analyses of mineral nutrition in calcareous soils

Please add a reference here

The discovery that structural and defensive traits do not vary with soil is consistent with a previous study in New Zealand's lowland temperate rain forests (Wright et al., 2010). That study compared traits of trees growing on phosphorus rich alluvium versus phosphorus-depleted marine terraces. Foliar phosphorus concentrations of species were halved on the marine terraces, but there was no detectable variation in structural traits, phenolic or tannin concentrations.

I would add more discussion at line 298. At the moment is more a description of results. Please specify at the beginning which traits are you talking about and why they do not change between poor and rich soils:

"Water" was defined as trait. Please define exactly what do you mean with water and how this was computed also here

Line 304: Species had a greater influence on trait values than soils for all traits, except P.

This makes completely sense to me because the content of P in leaves should be more related to the P available in the soil for the plants and not too much to the species. But again I found the discussion poor. There is a lot of literature about the leaf stoichiometry and P stoichiometry and the relationship with physical and chemical properties of the soil. Also with the database the authors have they can also explore how the reflectance is related to ration such as C/P N/P or C/N ratios.

Line 350 The region of importance with correlated wavelengths with nitrogen varies between 1192 nm in deciduous forest (Bolster et al., 1996) to 2490 for forage matter (Marten et al., 1983), which results directly from nitrogen in the molecular structure.

Please also cite other recent papers showing similar results with spectrometers similar

to the one used in this study (e.g. Homolova et al., 2013).

Line 353 Although chlorophylls also contain nitrogen, the spectra of chlorophylls differ greatly from proteins because of their dissimilar chemical structures, showing strong absorption due to C-H bonds in the phytol tail of the molecule (Katz et al., 1966),

Here if I understand correctly the authors they want to make the point that Chl and N are estimated with different regions of the spectrum despite N is one component of Chl and should covary. If my interpretation is correct I suggest another line of argumentation: Nitrogen Chl are contained in the green vegetation and N content and Chl are correlated (see Houborg et al., 2013). However, in dry leaves there is only N and not Chl. And therefore we cannot expect that the PLSR select similar regions for Chl and N.

Homolova, L., Malenovsky, Z., Clevers, J.G.P.W., García-Santos, G., Schaepman, M.E. Review of optical-based remote sensing for plant trait mapping (2013) Ecological Complexity, 15, pp. 1-16.

Houborg, R., Cescatti, A., Migliavacca, M., Kustas, W.P. Satellite retrievals of leaf chlorophyll and photosynthetic capacity for improved modeling of GPP (2013) Agricultural and Forest Meteorology, 117 (1), pp. 10-23.

---

## Author Comment (AC1) · 3 Feb 2017

Dear Dr. Michael Bahn, Thank you for your email regarding manuscript Leaf trait variation and field spectroscopy of generalist tree species on contrasting soil types. We were pleased that Referee #1 made positive comments. We are grateful that the reviewer went through the text carefully and spotted numerous minor issues, all of which we have resolved. We have improved the Material and Methods section to make it clearer, have supported with relevant literature the claims and statements as suggested by the reviewer and clarified both major and minor points. We have reduced the discussion on Si and broadened the review out to include other traits.

Yours sincerely Matheus Henrique Nunes

  Response to Anonymous Referee #1's comments

General Comments :

Referee comment: This is a well written, interesting paper that attempts to use leaf spectroscopy to predict leaf traits in two contrasting soil types. They found that traits tended to be specific to species and that soil type had much less of an influence. They used the PLSR methodology to predict traits with spectroscopy and found reasonably good relationships which reflect previous studies. Overall, this is a solid analysis and asks a relevant question of interest to the readers of this journal. Author response: We thank the referee for these positive comments.

Referee comment: Below I suggest a few areas where the paper could be strengthened and a number of minor points. Specific comments:

Referee comment: 1) The Material and Methods 'Statistical Analyses' section needs to be much expanded and clarified. Especially in regards to Figures 3 and 4. Without knowing how the data for those sections were acquired, it is difficult to evaluate the claims made in the results and discussion section.

Author response: We have expanded the text in the methods section to clarify how we acquired data and statistical analysis. Particularly, we added the sentence to clarify the Fig.3 "To evaluate the correlation among traits, Spearman correlation coefficient was calculated between all trait pairs and the variables were organised in the graphic using hierarchical clustering order to arrange traits into groups." The necessity of the fourth question was questioned (see below) given its similarity to the third question. In response, we have merged them into only: "What traits can be accurately and precisely estimated using spectroscopy in temperate woodlands?" We decided that Figure 4 was no longer necessary as it didn't not provide additional insights over the results provided elsewhere in the paper.

Referee comment: 2) Some of the findings discussed in the abstract need to be made clearer.

Author response: We have added more references in the discussion that support our findings. Particularly, we have substantially improved the "Phenotypic variation associated with soil" section, by discussing how P deficiency could be associated with variation in leaf traits. Furthermore, we have improved the "Inter-specific and residual variation" section in the discussion.

Referee comment: 3) Some of the claims/statements made in the abstract and intro either need to be changed or better supported with relevant literature.

Author response: We removed two questions that were initially part of the paper. We have decided that rather than force our traits into the three functional groups, we should run a single PCA to discover how the traits were related to each other, and to both species identity and soil type. Furthermore, we agree that the fourth question was similar to question 3, and have removed it. This gives us more opportunity to discuss the literature focusing on the two questions we have remaining. We changed some claims and statements in both introduction and discussion as highlighted by the referees.

Referee comment: 4) Many frequently used terms throughout the paper need to be changed/clarified (see below).

Author response: Many thanks for the suggestions on terminologies. It was done throughout the document, see comments below in minor amendments.

Referee comment: 5) Why there is such an emphasis on being able to predict Si using PLSR throughout the paper needs to be clarified!

Author response: We have reduced the discussion on Si and broadened the review out to include other traits.

Referee comment: 6) Discuss more clearly the relevance of the findings in terms of future high resolution aircraft campaigns. Based on these results, what sort of aircraft

data could be produced for temperate forests.

Author response: These points are clarified, changed or added to the manuscript. See below for more details.

Referee comment: Technical Corrections Terminology. Change uses of "among species' to "between species" (if that is what is meant). Change uses "species x soil interaction" to "species-soil interaction" or something similar. Change uses of "goodness-of-fit" to "strength of relationships" or something similar. Change uses of "leaf property" to "leaf trait". Line 58 – typo. Change "include phosphorous" to "including phosphorus". Line 84 – "leaf property". Replace with "leaf trait"? Line 108: "Leaf areas were measured". Suggested "Leaf area was measured?" Line 169: "strong co-llinearity". Typo. Line 326: "relative". Typo (relatively) Line 357: "A review in the literature". "A review of the literature"

Author response: Thank you for pointing out these numerous issues. We have made the corrections throughout the paper.

Referee comment: Abstract/summary Line 10 – change "traits variation" to "trait variation"

Author response: We now use the term "trait variation" and, in some cases, "variation in traits".

Referee comment: Line 12 – "Hyperspectroscopy is a recently developed technology for estimating the traits of fresh leaves" – disagree (the technique dates back to the 90s – e.g. Curran, 1989)

Author response: The claim was wrong indeed. We have replaced it with a sentence highlighting the importance of hyperspectroscopy for vegetation science. "There is currently great interest in hyperspectroscopy in vegetation science, particularly because improved airborne sensors and faster computing make it possible to map functional traits from the air (Asner and Martin, 2016; Jetz et al., 2016; Asner et al., 2017)".

Referee comment: Line 13 – "Few studies have evaluated its potential for assessing inter- and intraspecific trait variability in community ecology" – This is a contentious claim given a lot of Asner's work (e.g. Asner and Martin, 2011). This statement is not supported in the introduction.

Author response: We are not assessing directly the use of spectroscopy to predict inter- and inter-specific variance anymore because this question was overlapping with the third question. However, we agree that there are numerous works that evaluate the variation of a large range of traits in tropical forests (e.g. Asner's studies), however few studies have done in temperate forests for a large suite of traits.

Referee comment: Line 14 – "Working with 24 leaf traits". Contradicted by line 151 which mentions 26 leaf traits.

Author response: The number is 24 and we have altered the text accordingly.

Referee comment: Line 19 - "(iv) Can leaf spectra be used to detect inter-soil as well as inter-specific variation in traits?" – I don't understand how this question differs from iii ("what traits can be estimated precisely using field spectroscopy?"). If you can precisely estimate a trait using field spectroscopy, then surely it will be able to detect inter-soil and inter-specific variations. Unless the estimation only works on one species type on one particular soil type. Maybe rephrase?

Author response: We agree with you that this question do not differ from question iii. We decided to leave it out to make the paper more concise and focus on the question iii.

Referee comment: Line 20 – "The contribution of species and soil-type effects to variation in traits were evaluated using statistical analyses" – maybe state a few of the main statistical analyses used?

Author response: Thanks for the comment. We agree that it should be better clarified in the abstract. We changed it to "The contribution of species identity and soil-type

effects to variation in traits were evaluated using analysis of variance (ANOVA). Foliar traits were predicted from spectral reflectance using partial least square regression."

Referee comment: Line 21 – "Foliar traits were predicted from spectral reflectance using partial least square regression, and so inter- and intra-specific variation." – Presumed typo – rewrite.

Author response: We have changed the text to: Foliar traits were predicted from spectral reflectance using partial least square regression.

Referee comment: Line 22 – "Most leaf traits varied greatly among species" – a) replace 'among species' with either within or between species (presumed between?) b) Also this sentence is confusing – suggests that there was simply a wide variation in leaf trait measurements - slightly random to mention in abstract. Actual meaning I think is something along these lines "Inter-specific variation was the largest contributor to trait variation".

Author response: We have altered the sentence to: "Foliar traits were predicted from spectral reflectance using partial least square regression. inter-specific variation was the largest contributor accounting for 25% of the total variation in all leaf traits"

Referee comment: Line 23 – "Macronutrient concentrations were greater on alluvial than chalk soils while micronutrient concentration showed the opposite trend" – Foliar macronutrient concentrations or soil macronutrient concentrations? (presumed the former?). Also, slightly odd sentence – what's the significance? Maybe meant to say something along these lines? - "However, foliar macro- and micronutrient concentrations were found to be more strongly influenced by soil type".

Author response: We have changed the text to: Overall, inter-specific variation was the largest contributor accounting for 25% of the total variation in all leaf traits and soil type accounted for 5%, and the interaction between species and soil accounted for virtually no variation. In general, plants investing in traits related to growth had less investment

in defence or foliar structure, whereas soil played minor important role in these traits. Nonetheless, foliar macro- and micronutrient concentrations were found to be strongly influenced by soil type, and foliar phosphorus had the largest variation among all traits due to soil variation.

Referee comment: Line 24 – "Si predictions using spectroscopy appear to be promising" – what's so special about Si predictions?! Why do they get singled out?

Author response: It was the first time Si was reported as a trait able to be predicted using spectroscopy in forests. But we agree that it should not be singled out as Si is not the main focus for the paper. We have rephrased the sentences that were mentioning Si as a very important finding and we have reduced the amount of text on Si in the discussion.

Referee comment: Line 28 – "However, it [field spectroscopy] was unable to detect subtle within species variation of traits associated with soil type" – repetition of line 25? ("Field spectroscopy. . ..was less effective at detecting subtle variation of rock-derived nutrients between soil types"). Combine sentences to keep abstract concise?

Author response: We are not assessing directly the ability of spectroscopy to detect variation of traits associated with soil type, so we have removed the sentences related to it.

Introduction

Referee comment: Line 64 – "along environmental change". Typo. Suggested "along environmental gradients"?

Author response: We have changed the text to "In response to environmental change".

Referee comment: Line 71 – "However, spectral and chemical properties may be uncoupled if intraspecific variation in foliar traits is high and/or phenotypic plasticity exceeds phylogenetic patterns among leaf properties". Disagree. Spectral and chemical relationships would still hold, it would just be harder to identify species type based on

their reflectance signatures.

Author response: We agree with you. We found references supporting your statement: "Wu et al., (2016) found that leaf traits and spectra vary systematically and predictably with leaf age between forest sites on contrasting soil types thousands of kilometers apart." Madritch et al. (2014) also demonstrated the high correlation between spectral properties, traits and soil very well. We have changed the text and added the reference: "...structural differences (i.e., leaf thickness, number of air water interfaces, cuticle thickness, and pubescence) between leaves may have significant effects on the relationship between leaf reflectance and traits (Sims and Gamon, 2002)."

Madritch, M.D., Kingdon, C.C., Singh, A., Mock, K.E., Lindroth, R.L. and Townsend, P.A., 2014. Imaging spectroscopy links aspen genotype with below-ground processes at landscape scales. Philosophical Transactions of the Royal Society of London B: Biological Sciences, 369(1643), p.20130194.

Wu, J., Chavana‐Bryant, C., Prohaska, N., Serbin, S.P., Guan, K., Albert, L.P., Yang, X., Leeuwen, W.J., Garnello, A.J., Martins, G. and Malhi, Y., 2016. Convergence in relationships between leaf traits, spectra and age across diverse canopy environments and two contrasting tropical forests. New Phytologist.

Referee comment: Line 73 – "Martin and Aber (1996) demonstrated that equations for estimating leaf properties from one site were unable to predict leaf properties for other sites, due to variability in the magnitudes of foliar traits levels between data sets and environmental influences". Very old reference and what about all the evidence to the contrary (e.g. all of Asner's work) ???

Author response: As per previous comment, we have deleted that reference. We also agree that we can find high correlation between spectral properties, traits and soil (see previous comment).

Referee comment: Line 75 – "To our knowledge, the link between foliar traits and

spectral properties of trees has not been broadly demonstrated for temperate forests"
– query this statement. The remote sensing of foliar traits began in temperate forests.

Author response: We rephrased that statement showing that temperate forests have
been studied to : To our knowledge, the link between foliar traits and spectral properties
of trees has not been broadly demonstrated for a large range of traits on contrasting
soil types in temperate forests. There is currently great interest in hyperspectroscopy
in vegetation science, particularly because improved airborne sensors and faster com-
puting make it possible to map functional traits from the air (Asner and Martin, 2016;
Jetz et al., 2016; Asner et al., 2017)

Referee comment: Line 86 – "what is the relative contribution of soil type and species
to leaf trait variation?". Missed word? "what is the relative contribution of soil type and
species type to leaf trait variation".

Author response: We have included the word species identity.

Referee comment: Line 88 – "does the importance of the three functional groups
change due to soil or more due to species variation?" – awkward phrasing. Rephrase.

Author response: We recognise that grouping functional classes is controversial. We
decided to eliminate this question as it was in fact confusing and somewhat misleading.

Material and Methods Referee comment: Line 102: "Leaves of 66 trees of six species
were collected from the two contrasting soil types. The six species were in common
to both sites". Suggested "Across both sites, leaves were collected from 66 trees,
representing six species. The six species common to both sites were:"

Author response: Many thanks. We have made this amendment.

Referee comment: Line 103: "Acer campestre L. (Field Maple)" – what does the L
stand for?

Author response: L. is the authority - the person who named the species formally.

[Figure]

In this case Linnaeus, who back in the 1700s invented the Latin binomial system for naming species that is still used today. Some biology journals insist on including these. We have removed them from this paper

Referee comment: Line 105: "Two fully sunlit branches were selected, were cut and placed on ice in a cool box, and transported to a lab for processing within 2 hours (and often within 30 minutes)".

Author response: We have changed the text to: Two fully sunlit branches were selected, cut and placed in a cool box, and subsequently transported to a laboratory for processing within two hours.

Referee comment: Line 149: "2.4 Statistical analyses". Needs to be split up into each statistical analysis performed and titled accordingly.

Author response: This has been carried out as requested.

Referee comment: Line 156: "Where necessary, variables were log transformed to meet assumptions of ANOVA".

Author response: Table 1 has additional information concerning which variables were log transformed and how they can be found.

Referee comment: Line 168: PLSR section – no mention of using 70% to calibrate and 30% to test but Cal and Val appear on Table 3. No mention of how the data for Figure 3 and 4 is acquired!!!

Author response: We added the following sentences to the text to make it clearer: "We adopted a leave-one-out cross-validation for each PLSR model. Model accuracy was expressed by the coefficient of determination ($R^2$) and root mean square error (RMSE). We also standardised RMSE to the percentage of the response range (RMSE%) by dividing each RMSE by the maximum and minimum values of each leaf trait ?? what?? , as in Feilhauer et al., 2010. RMSE and $R^2$ were acquired during both model calibration (indicated with subscript cal) and after model validation (indicated as subscript

val)." Regarding Figure 3: To evaluate the correlation among traits, Spearman correlation coefficient was calculated between all trait pairs and the variables were organised in the graphic with hierarchical clustering order. Regarding Figure 4: We deleted question iv and, subsequently, Figure 4.

Results Referee comment: Line 204 – "Species exerted little or no influence on pigment concentrations" – Refer to species in this context (and throughout paper) as 'species type"?

Author response: Yes. We have changed this throughout the document to species identity.

Referee comment: Line 241: "Ability to predict leaf traits from hyperspectral reflectance varied greatly among the 24 traits fitted using the 6 species (Table 3)". "fitted using the 6 species" - confusing. Rephrase.

Author response: We removed the "fitted using the 6 species" as it did not make sense.

Referee comment: Line 243: "PLSR modelling for LMA, water, Si, phenolics, carotenoids, K, B, efficiency of PSII, N, chlorophyll a and chlorophyll b were in descending order the best performing in terms of"

Author response: Thank you. We have corrected the text.

Referee comment: Line 248- "higher goodness-of-fit" – use a different term? E.g. stronger relationships/correlations etc.

Author response: OK – we have changed the text to higher strength of relationship.

Referee comment: Line 256: "There were strong correlations among some of the leaf properties (Fig. 3) that can be potentially leveraging the estimation of other leaf traits from the use of PLSR". Interesting. Explain further?

Author response: We added the following sentences in the results: There were strong correlations among some of the leaf traits that may have leveraged the estimation of

other leaf traits from the use of PLSR. Si and B were highly correlated to hemicellulose, cellulose and lignin, for instance, but spectroscopy sensitivity to those traits is an artefact of traits correlation rather than a real feature. Likewise, K was highly correlated to leaf water content, soluble carbons, lignin, hemicellulose and cellulose, and its strong relationship with the spectral information can also represent the artefact of correlation with other traits directly predicted using spectroscopy.

Referee comment: Line 257: "The correlation graphic also shows the similarity among variables through cluster analysis". Explain. Cluster analysis was not been mentioned in the Materials and Methods. Explain how this was achieved, why it was done and expand on results.

Author response: We have made it clearer in the Material and methods as follows: To evaluate the correlation among traits, Spearman correlation coefficient was calculated between all trait pairs and the variables were organised in the graphic with hierarchical clustering order.

Discussion

Referee comment: Line 271: "Some leaf traits were strongly influenced by both species and soil type, while others were hardly affected by soil and only varied with species". Vague. Make more specific.

Author response: We have been more specific and have changed the text to: Some leaf traits were greatly influenced by species identity with influences of soil type, such as soluble carbons, N, K, Mg, B and Zn, whereas other traits were hardly affected by soil and only varied with species, such as cellulose, hemicelluloses, lignin, foliar water content, Si and phenolics. Soil had a strong influence on concentrations of mineral nutrients in the leaves. Other foliar traits – mostly those involved in structure, defence and growth - varied between species but soil had little detectable effect.

Referee comment: Line 305: "water" – change to 'leaf water content'.

Author response: Thank you. We have done all the corrections ad changed the term "water" to "leaf water content" throughout the text.

Referee comment: Line 321: "but their study sampled only from fully sunlit leaves". Suggested - "Similarly, their study sampled only from fully sunlit leaves".

Author response: Thank you. Alteration made.

Referee comment: Line 325: "The investment in light capture had high intra-specific variation, and neither species nor soil accounted for variation in [these] foliar properties". Missing word.

Author response: We have restructured the sentence.

Referee comment: Line 327: "separating out some species". Confusing. Rephrase?

Author response: We have improved the discussion and included the following sentence that had the same meaning: "The investment in structure and defence-related traits were little influenced by soil type and was mainly determined by species identity."

Referee comment: Line 327: "Investment in traits related to defence and leaf structure is species-mediated, and may be separated into two defensive strategies". State the two defensive strategies?

Author response: We improved the discussion regarding the functional grouping as follows: "Species identity is driving investment in defence and foliar structure, as well as investment in traits related to growth. In general, our study indicates that plants investing in compounds related to defensive strategies (e.g. either high phenolics or high carbon fractions concentrations) are associated with plants investing less in traits related to growth (e.g. P, N, water content, soluble carbons), although traits are not necessarily individually correlated to each other."

Referee comment: Line 342: "Doing so revealed that. . .". Awkward. Rephrase.

Author response: This sentence does not exist anymore after the changes throughout

the discussion.

Referee comment: Line 351: "Although chlorophylls also contain nitrogen, the spectra of chlorophylls differ greatly from proteins because of their dissimilar chemical structures, showing strong absorption due to C-H bonds in the phytol tail of the molecule (Katz et al., 1966), also confirmed in this work when visualizing the regions of importance for predictions." Require a full stop after (Katz et al. 1996) and develop last sentence ("also confirmed in this work when visualizing the regions of importance for predictions").

Author response: We have refined the text to: Although chlorophylls also contain nitrogen, the spectra of chlorophylls differ greatly from proteins because of their dissimilar chemical structures, showing strong absorption due to C-H bonds in the phytol tail of the molecule (Katz et al., 1966). That can be confirmed in this work as the visible region of the spectrum showed the best predictions of pigments.

Referee comment: Line 360: "On the other hand, the use of spectroscopy on fresh leaves is particularly better for LMA predictions"

Author response: We have edited the paragraph, which included that specific sentence to : "The use of spectroscopy on fresh leaves is particularly positive for LMA predictions."

Referee comment: Line 365: "The use of spectroscopy for Si predictions on fresh leaves appears to be promising considering our accurate results". Maybe, but why are Si predictions so important? What ecological function does Si perform?!

Author response: We have reduced the text on Si to avoid singling it out, as it is not a specific question on this paper.

Referee comment: Line 339: 4.4 Predictions of foliar traits using spectroscopy – this section maybe a bit long? Could condense? Says some interesting things but I'm not sure they're all relevant to the paper.

Author response: We have condensed the whole section 4.4 as requested

Referee comment: Line 384: Consideration on the use of spectroscopy to quantify patterns of foliar traits. Typo - Consideration of the use of spectroscopy to quantify patterns of foliar traits.

Author response: Thanks. The correction was made.

Referee comment: Line 385. "The range of variation within species for most predicted traits tend to be smaller with the use of PLSR on reflectance". Very confusing. Rephrase.

Author response: As we deleted the question iv off, this sentence is no longer on the paper.

Referee comment: Line 399: "This study particularly provides findings for a large range of traits that indicate that the use of spectroscopy may be useful to quantify structural traits but can be misleading to measure the environmental filtering on traits that are indirectly predicted, such as macro- and micronutrients". I might agree if I understood Figure 4 but, as I don't, I query this statement.

Author response: As we deleted the question iv off, this sentence is no longer on the paper.

Referee comment: Line 401: "While remote sensing is not a direct replacement of field sampling, the ability of remote sensing platforms to assess biological phenomena at large spatial scales is unparalleled". Slightly random – doesn't follow from previous statement/results section.

Author response: We agree that it completely disagree with previous statements and results. We have changed to: "This study demonstrates the potential for rapid and accurate estimation of foliar traits. Soil fertility is not constraining the use of spectroscopy to a particular soil type, indicated by the strong relationship between foliar traits and spectral information across both soils. Spectroscopy provides the opportunity to characterize important sources of variation in foliar traits related to chemistry without having to measure the entire range of foliar constituents. Our results provide the basis for additional studies to use spectroscopy to identify additional foliar constituents that may vary among temperate forests."

Conclusion

Referee comment: Line 407: "rock-derived nutrients are strongly influenced by the soil characteristics". Need to tone down or change previous sentence, otherwise statements are contradictory.

Author response: We have changed to: "Species identity had a much stronger influence on most traits than the substrate upon which the trees grew. In particular, most traits associated with light capture, growth, cell wall structure and defence were uninfluenced by substrate. The exception to this rule, was that foliar concentrations of rock-derived nutrients were strongly affected by the soil type. Plants investing into defence or foliar structure are investing less in growth, patterns greatly influenced by species identity and, with much less influence, by soil type. "

Referee comment: Line 409: "This study also demonstrates the potential for estimating foliar traits by field spectroscopy and its promising use to predict Si". a) "demonstrates the potential" –this has already been done many times. Maybe something more along the lines of "agrees with the existing literature in demonstrating the potential. . ." b) "its promising use to predict Si". Once again – what is so important about Si?!?!?!

Author response: We have relatively changed the discussion to make it clearer and subsequently the conclusion. That sentence is now as follows: "Some traits, for instance Si, B and K, are likely to be accurately predicted due to the integrating information on several foliar traits simultaneously. This study demonstrates the potential for rapid and accurate estimation of foliar traits of forest canopies with contrasting soil types. Soil fertility is not constraining the use of spectroscopy to a particular soil type, indicated by the strong relationship between foliar traits and spectral information across

both soils."

Figures

Referee comment: Line 661: "Red and black circles mean negative and positive correlations". Which way round?

Author response: The figure is no longer on the paper as question iv was deleted.

Referee comment: Line 668: "The greyness and size of each dot reflects the goodness-of-fit of the PLSR for each foliar trait, with darker and bigger points representing the most accurate PLSR predictions. goodness-of-fit". Give statistical boundaries for how dots were sorted into each size/shape category.

Author response: The figure is no longer on the paper as question iv was deleted.

Referee comment: Perhaps add the word "...respectively" at the end to clarify which is which?

Author response: Thank you. The correction was made.

Referee comment: Line 675: Table 1. CV needs to be represented as %CV, as stated in the heading. Author response: Thank you. It is corrected now.

Please also note the supplement to this comment:
http://www.biogeosciences-discuss.net/bg-2016-432/bg-2016-432-AC1-supplement.pdf

———————————————

---

## Author Comment (AC2) · 3 Feb 2017

Dear Dr. Michael Bahn, Thank you for your email regarding manuscript Leaf trait variation and field spectroscopy of generalist tree species on contrasting soil types. We were pleased that Referee #2 saw merit in our work. We are grateful that the reviewer went through the text carefully and gave us positive ideas on variation in traits, all of which we have resolved. We have substantially improved the discussion, have included a figure that includes the reflectance spectra along with an indication of the regions relevant to estimate each leaf trait (Figure 1) and have dealt with functional grouping from a different perspective. We also have reduced the discussion on Si and broadened the review out to include other traits.

[Figure]

Yours sincerely Matheus Henrique Nunes

Response to Anonymous Referee #2's comments

Referee comment: The article "Leaf trait variation and field spectroscopy of generalist tree species on contrasting soil types" by Nunes and co-authors analyzed field spectroscopy data collected on different European tree species on contrasting soil types. The authors worked with 24 leaf traits and explored the following questions: What contribution do soil type and species identity make to trait variation? When traits are clustered into three functional groups (light capture and growth, leaf structure and defence, as well as rock-derived nutrients), are some groups more affected by soil than others? What traits can be estimated precisely using field spectroscopy? Can leaf spectra be used to detect inter-soil as well as inter-specific variation in traits? The authors found that most leaf traits varied greatly among species. The effects of soil type were generally weak by comparison Specific Comments:

Referee comment: Line 28 variation in foliar traits and Si predictions using spectroscopy appear to be promising. Not clear what Si means at this stage, it becomes clear later. But in general all the discussion on Si is poor

Author response: Firstly, we spelled out Si and all the nutrients that were presented on the paper as an acronym. We previously singled out the performance of Si as a promising result but its performance should not be the main focus of the manuscript. We have reduced the discussion on Si and broadened the review out to include other traits.

Referee comment: Line 162 We recognize that grouping leaf properties into functional classes can be controversial, given that a single leaf property can contribute to This is particularly true for P, this assumption has to be justified as foliar P can be easily considered a trait associated to growth.

Author response: We recognise that grouping leaf traits into functional classes can be controversial, given that a single leaf trait can contribute to more than one class (e.g. LMA is related to growth but also to defence, P is a rock-derived nutrient also associated with growth). We based our leaf traits grouping on previous studies that attempted to investigate this chemical portfolio that expresses multiple strategies undertaken by plants to maximize fitness over the lifetime of the individual or species (Asner and Martin, 2012; Asner et al., 2015). Furthermore, we have deleted the second question where we attempt to model variation within each pre-determined group, but we use principal component analysis to group all traits based on the data instead to see whether our traits follow those groupings. Asner, G.P. and Martin, R.E., 2012. Contrasting leaf chemical traits in tropical lianas and trees: implications for future forest composition. Ecology Letters, 15(9), pp.1001-1007. Asner, G.P., Anderson, C.B., Martin, R.E., Tupayachi, R., Knapp, D.E. and Sinca, F., 2015. Landscape biogeochemistry reflected in shifting distributions of chemical traits in the Amazon forest canopy. Nature Geoscience, 8(7), pp.567-573.

Results Section Spectroscopy of leaf properties

Referee comment: he results of PLSR are on one hand encouraging because the portion of spectra selected for specific traits are in line with what expected from the literature. Some examples from the article: 1) higher goodness-of-fit were obtained for K, Ca and P in the SWIR regions. 2) Pigments were the only traits that predictions were more accurate when using the visible region (400 – 700 nm)

Author response: Many thanks. We thought this encouraging too.

Referee comment: I think would be useful to have more discussion on what is known and what is new compared for instance to the review from Homolova et al., which discuss many of the traits mentioned by the authors and how these traits can be predicted from remote sensing data. What do we learn from these results? I think the authors should make an effort to improve this aspect because can be quite relevant considering

the great dataset they have. For example a figure with a reflectance spectra with an indication of the regions relevant to estimate other the traits indicated might be useful for the reader.

Author response: I agree that it would be interesting to have a figure with an average reflectance spectrum indicating the relevant regions for each trait, as per Figure 1. We included the Coefficient of variation (%) and the average reflectance with the regions partitioning indicating which part of the spectrum is more suitable for each trait. There are amendments in the Material and Methods, as well as Results sections on the graphic. Figure 1. Spectral reflectance and coefficient of variation (% CV) of reflectance of six generalists species for alluvial and chalk soils. The spectral regions for each trait were selected based on the model that minimised RMSE.

Referee comment: Line 267 The species x soil interaction effects were detected by PLSR modelling, except for traits that showed strong interaction (Mn, P and $\delta$ 13C). This should be better discussed

Author response: We thought that the fourth question could be leading to question iii, and was irrelevant to bring more information into the paper. Thus, the abovementioned sentenced is no longer on the manuscript.

Referee comment: Line 279 Our findings that trees growing on the chalk soils had relatively low concentrations of N, P and K in their leaves, and relatively high concentrations of Ca, Mg, B, Mn, Si and Zn, is consistent with previous analyses of mineral nutrition in calcareous soils. Please add a reference here

Author response: Our findings that trees growing on the chalk soils had relatively low concentrations of N, P and K in their leaves, and relatively high concentrations of Ca, Mg, B, Mn, Si and Zn, is consistent with analyses of foliar nutrients in chalk grasslands species by Hillier et al. (1990). Thin chalk soils contain small quantities of macronutrients needed by plants, and are unproductive for growing crops unless heavily fertilized; however, cation exchange sites in the soil contain high concentrations of calcium and

magnesium (Hillier et al., 1990).

Referee comment: The discovery that structural and defensive traits do not vary with soil is consistent with a previous study in New Zealand's lowland temperate rain forests (Wright et al., 2010). That study compared traits of trees growing on phosphorus rich alluvium versus phosphorus-depleted marine terraces. Foliar phosphorus concentrations of species were halved on the marine terraces, but there was no detectable variation in structural traits, phenolic or tannin concentrations. I would add more discussion at line 298. At the moment is more a description of results. Please specify at the beginning which traits are you talking about and why they do not change between poor and rich soils:

Author response: We have added more references and made the sentences clearer: "The investment in structure and defence-related traits were little influenced by soil type and was mainly determined by species identity. The discovery that structural and defensive traits (i.e. lignin, phenolics) do not vary with soil is consistent with a previous study in New Zealand's lowland temperate rain forests (Wright et al., 2010). The authors compared traits of trees growing on phosphorus-rich alluvium versus phosphorus-depleted marine terraces, and found that concentrations of these compounds were invariant (see also Koricheva et al. 1998; Long et al. 2016). LMA does not vary with soil type and did not correlate with nutrients in the leaves. High LMA, however, is associated with higher pigments in the leaves and, therefore, pigments play a role in modulating LMA variability. The effect of low nutrient availability on leaf anatomy is much smaller than the effect of light (Shields, 1950) and, consequently, the overall effect of nutrients on LMA is moderate, and (on average) only appears when plants are severely limited in growth (Poorter et al., 2009). In general, the concentration of rock-derived nutrients in leaves is highly dependent on soil type as environmental filter. Traits favouring high photosynthetic rate and growth are considered to be advantageous in rich-resource soil environments, whereas expressions of traits favouring resource conservation are considered advantageous in low-resource environments

(Aerts and Chapin, 1999, Westoby et al., 2002).

Referee comment: "Water" was defined as trait. Please define exactly what do you mean with water and how this was computed also here

Author response: We included the following sentence on the paper: "Leaf water content was computed as the ratio between the quantity of water (fresh weight – dry weight) and the fresh weight." We also used the term leaf water content throughout the paper.

Referee comment: Line 304: Species had a greater influence on trait values than soils for all traits, except P. This makes completely sense to me because the content of P in leaves should be more related to the P available in the soil for the plants and not too much to the species. But again I found the discussion poor. There is a lot of literature about the leaf stoichiometry and P stoichiometry and the relationship with physical and chemical properties of the soil.

Author response: We agree that some discussion on P was missing out. We have included some sentences on P and the relationship with other variables along the discussion: Leaf P is related to soil P, which not necessarily affects foliar N (Ordoñez et al., 2009), however the effect of soil P on leaf N seems determined by a tight coupling of leaf N and leaf P (Niklas et al., 2005).

Referee comment: Also with the database the authors have they can also explore how the reflectance is related to ratio such as C/P N/P or C/N ratios.

Author response: We did not obtain a strong relationship between P and spectral data, which can be attributed to the low P concentration in the leaves (Homolova et al., 2013). According to these authors, there is a limited number of studies that estimated P using spectroscopy revealing inconsistent spectral bands among the reviewed literature. P is poorly predicted with field spectroscopy, and so did tested ratios including P, and for this reason we decided against evaluating stoichiometry in this paper, interesting though it is. As P predictions using spectroscopy might be an artefact of correlation

with other traits, we decided not to include ratios that would not be directly detected.

Referee comment: Line 350 The region of importance with correlated wavelengths with nitrogen varies between 1192 nm in deciduous forest (Bolster et al., 1996) to 2490 for forage matter (Marten et al., 1983), which results directly from nitrogen in the molecular structure. Please also cite other recent papers showing similar results with spectrometers similar to the one used in this study (e.g. Homolova et al., 2013).

Author response: Thank you for the suggestion. We have included it: "According to Kumar et al. (2001), three main protein absorption features report as important for N estimation are located around 1680 nm, 2050 nm and 2170 nm."

Referee comment: Line 353 Although chlorophylls also contain nitrogen, the spectra of chlorophylls differ greatly from proteins because of their dissimilar chemical structures, showing strong absorption due to C-H bonds in the phytol tail of the molecule (Katz et al., 1966), Here if I understand correctly the authors they want to make the point that Chl and N are estimated with different regions of the spectrum despite N is one component of Chl and should covary. If my interpretation is correct I suggest another line of argumentation: Nitrogen Chl are contained in the green vegetation and N content and Chl are correlated (see Houborg et al., 2013). However, in dry leaves there is only N and not Chl. And therefore we cannot expect that the PLSR select similar regions for Chl and N.

Author response: The region of importance with correlated wavelengths with nitrogen varies between 1192 nm in deciduous forest (Bolster et al., 1996) to 2490 for forage matter (Marten et al., 1983), which results directly from nitrogen in the molecular structure. According to Kumar et al. (2001), three main protein absorption features report as important for N estimation are located around 1680 nm, 2050 nm and 2170 nm. Although chlorophylls also contain nitrogen, the spectra of chlorophylls differ greatly from proteins because of their dissimilar chemical structures, showing strong absorption due to C-H bonds in the phytol tail of the molecule (Katz et al., 1966). That can be con-
firmed in this work as the visible region of the spectrum showed the best predictions of pigments. Chl and N were not correlated in our study and the spectral measurements were done on fresh leaves. The main reason for PLSR to select different regions was that N is correlated to the proteins and Chl (even though they contain nitrogen) to the phytoil tails.

Homolova, L., Malenovsky, Z., Clevers, J.G.P.W., García-Santos, G., Schaepman, M.E. Review of optical-based remote sensing for plant trait mapping (2013) Ecological Complexity, 15, pp. 1-16. Houborg, R., Cescatti, A., Migliavacca, M., Kustas, W.P. Satellite retrievals of leaf chlorophyll and photosynthetic capacity for improved modeling of GPP (2013) Agricultural and Forest Meteorology, 117 (1), pp. 10-23.

Please also note the supplement to this comment: http://www.biogeosciences-discuss.net/bg-2016-432/bg-2016-432-AC2-supplement.pdf

―――――――――――――――――

[Figure]

**Fig. 1.**

---

## Author Response (AR1)

Dear Dr Bahn,

Thank you for your email regarding manuscript *Leaf trait variation and field spectroscopy of generalist tree species on contrasting soil types.* We are grateful to reviewers for their very thorough review. We were pleased that referee #1 regarded the manuscript as "a well written, interesting paper" based on "a solid analysis" which "asks a relevant question of interest to the readers of this journal". However, the review proceeded to identify many typographical errors and points that needs clarification. Referee #2 explained that they were unable to understand two of our analyses and the associated figures.

We have revised the text carefully following the reviewers' suggestions. Both referees criticised our choice to focus the discussion on the spectroscopy of foliar silicon; we now present a much broader perspective on the uses and limitations of field spectroscopy for detecting multiple traits. We have also sought to emphasize the key points of the paper by replacing two figures that the referees found difficult to understand with much simpler figures that convey the same message. As requested by reviewer #2, we have included a figure that shows the reflectance spectra along with an indication of the regions relevant to estimation of different leaf traits (Figure 1) and have removed the sections on functional groupings. Also, as requested by reviewer #2, P supply limitation was better discussed, as well as the soil and species effects on traits.

We now focus the paper around one issue: the challenges of measuring intraspecific variation in some leaf traits using field spectroscopy. Rock-derived nutrients lack absorption features in visible to shortwave-infrared region of the electromagnetic spectrum so cannot be measured directly by spectroscopy. They can, nevertheless, be estimated indirectly because element concentrations co-vary with organic molecules that do have strong absorption features ("constellation effects"). Our paper identifies a problem with this approach: there were strong differences in rock-derived mineral nutrients between soil types, but we could not measure these because the concentrations of defence and structural traits (used to indirectly estimate the mineral nutrient concentrations) were barely affected by soil type. We have shown many similarities between our study and those in tropical forests, demonstrating that this problem is likely to be widespread.

You requested major revision of the manuscript. These revisions have resulted in many changes to the text, as you will note in the track-changed document following the responses to the reviewers on the same document. However, the underlying analyses are unchanged. We thank the referees again for their insightful comments, and hope they find our revisions satisfactory.

Yours sincerely

Matheus Henrique Nunes and David Coomes.

**Response to Anonymous Referee #1's comments**

**General Comments :**

Referee comment: This is a well written, interesting paper that attempts to use leaf spectroscopy to predict leaf traits in two contrasting soil types. They found that traits tended to be specific to species and that soil type had much less of an influence. They used the PLSR methodology to predict traits with spectroscopy and found reasonably good relationships which reflect previous studies. Overall, this is a solid analysis and asks a relevant question of interest to the readers of this journal.

Author response: We thank the referee for these positive comments.

Referee comment: Below I suggest a few areas where the paper could be strengthened and a number of minor points. **Specific comments:**

Referee comment: 1) The Material and Methods 'Statistical Analyses' section needs to be much expanded and clarified. Especially in regards to Figures 3 and 4. Without knowing how the data for those sections were acquired, it is difficult to evaluate the claims made in the results and discussion section.

Author response: We have removed the former figure 4 and replaced it with a new figure that can be more easily interpreted (current Figure 5). We also have expanded the text in the methods section to clarify how we acquired and analysed the data. In particular, we have added the following sentence to clarify the methods used to construct the correlation matrix graphic: "To evaluate the correlation among traits, Spearman rank correlation coefficient was calculated between all trait pairs and the variables were ordered in the figure by hierarchical clustering."

We have provided a clearer explanation for Figure 5: "To evaluate the effectiveness of field spectroscopy at measuring variation in traits related to soil type and species identity, we partitioned variance in model-predicted trait values using exactly the same approach as we used with lab-measured traits (i.e. first paragraph of methods)."

[Figure]

Figure 5. Partitioning of variance of foliar traits between species, soil, species-soil interaction and residual components for six generalist species found on both chalk and alluvial soils from predicted data. Residual variation arises from within-site intraspecific variation, micro-site variability, canopy selection but not measurement error variance, and is therefore smaller than for field measurements (Fig. 1). Predicted data were obtained from partial least square regression (PLSR).

Referee comment: 2) Some of the findings discussed in the abstract need to be made clearer.

Author response: We have added more references in the discussion that support our findings. Particularly, we have substantially improved the "Variation in traits between chalkland vs alluvial soils" section, by discussing how P deficiency could be associated with

variation in leaf traits. Furthermore, we have improved the "Inter-specific and residual variation" section in the discussion.

Referee comment: 3) Some of the claims/statements made in the abstract and intro either need to be changed or better supported with relevant literature.

Author response: We removed two questions that were initially part of the paper. We have decided that rather than force our traits into the three functional groups, we should run a single PCA to discover how the traits were related to each other, and to both species identity and soil type. Furthermore, we agree that the fourth question was similar to question 3, and have merged both questions into a single one: "is field spectroscopy effective at predicting phenotypic variation in leaf traits between soil types, as well as interspecific differences?". We changed some claims and statements in both introduction and discussion as highlighted by the referees.

Referee comment: 4) Many frequently used terms throughout the paper need to be changed/clarified (see below).

Author response: Many thanks for the suggestions on terminologies. It was done throughout the document, see comments below in minor amendments.

Referee comment: 5) Why there is such an emphasis on being able to predict Si using PLSR throughout the paper needs to be clarified!

Author response: We have reduced the discussion on Si and broadened the review out to include other traits.

Referee comment: 6) Discuss more clearly the relevance of the findings in terms of future high resolution aircraft campaigns. Based on these results, what sort of aircraft data could be produced for temperate forests.

Author response: We have added as the last section of the discussion the "Difficulties in measuring phenotypic variation by field spectroscopy and its implications for mapping

functional traits". We have included some considerations on the relevance of our findings in terms of high resolution aircraft campaigns.

Referee comment: Technical Corrections Terminology.

Change uses of "among species' to "between species" (if that is what is meant).

Change uses "species x soil interaction" to "species-soil interaction" or something similar.

Change uses of "goodness-of-fit" to "strength of relationships" or something similar. Change uses of "leaf property" to "leaf trait".

Line 58 – typo. Change "include phosphorous" to "including phosphorus".

Line 84 – "leaf property". Replace with "leaf trait"?

Line 108: "Leaf areas were measured". Suggested "Leaf area was measured?"

Line 169: "strong co-llinearity". Typo.

Line 326: "relative". Typo (relatively)

Line 357: "A review in the literature". "A review of the literature"

Author response: Thank you for pointing out these issues. We have made the corrections requested.

Referee comment: Abstract/summary Line 10 – change "traits variation" to "trait variation"

Author response: We now use the terms "trait variation" and "variation in traits".

Referee comment: Line 12 – "Hyperspectroscopy is a recently developed technology for estimating the traits of fresh leaves" – disagree (the technique dates back to the 90s – e.g. Curran, 1989)

Author response: The claim was wrong indeed. We have replaced it with a sentence highlighting the importance of hyperspectroscopy for vegetation science. "There is currently great interest in using hyper-spectroscopy as a tool for studying the chemical and structural traits of leaves, particularly because improved airborne sensors and faster computing make

it possible to map functional traits from the air (G. P. Asner et al., 2017; Gregory P. Asner & Martin, 2016b; Jetz et al., 2016; Ustin et al., 2009). Plans to put hyperspectral sensors into space (e.g. DRL plan to launch EnMAP in 2018; Guanter et al. 2015) will soon enable spectral response curves of vegetation communities to be assessed at the global scale.".

Referee comment: Line 13 – "Few studies have evaluated its potential for assessing inter- and intraspecific trait variability in community ecology" – This is a contentious claim given a lot of Asner's work (e.g. Asner and Martin, 2011). This statement is not supported in the introduction.

Author response: We agree that the Asner's team have published several papers on this topic for temperate forests. But we argue on the text that analyses involving this large suite of traits provide optimism to develop general, predictive rules in community ecology as we refine our understanding of which traits are varying in a given environment. There is also a need for broader testing of the mechanisms underlying interspecific variation in phenotypic plasticity across traits and environmental variables (e.g. Weiner, 2004; Funk et al., 2016) and how this variation ultimately informs species and community responses to environmental change (Funk et al., 2016).

Referee comment: Line 14 – "Working with 24 leaf traits". Contradicted by line 151 which mentions 26 leaf traits.

Author response: The number is 24 and we have altered the text accordingly.

Referee comment: Line 19 - "(iv) Can leaf spectra be used to detect inter-soil as well as inter-specific variation in traits?" – I don't understand how this question differs from iii ("what traits can be estimated precisely using field spectroscopy?"). If you can precisely estimate a trait using field spectroscopy, then surely it will be able to detect inter-soil and inter-specific variations. Unless the estimation only works on one species type on one particular soil type. Maybe rephrase?

Author response: We agree with you. We have rephrased and merged both questions into a single one: "is field spectroscopy effective at predicting phenotypic variation in leaf traits between soil types, as well as interspecific differences?".

Referee comment: Line 20 – "The contribution of species and soil-type effects to variation in traits were evaluated using statistical analyses" – maybe state a few of the main statistical analyses used?

Author response: Thanks for the comment. We agree that it should have been better explained in the abstract. We changed it to "Analyses were performed within the R statistics framework (R Team 2014). To evaluate the correlation among traits, Spearman rank correlation coefficient was calculated between all trait pairs and the variables were ordered in the figure by hierarchical clustering. Analyses of variance (ANOVA) were used to examine the influences of species identity and soil type on each of the 24 leaf traits. Species, soil and soil x species terms were included in the model, and the ratio of sum of squares of these terms versus the total sum of squares was used as an index of species- versus site-level variation. This partitioning of variance quantifies the variation between species, between soil types, the interaction between soil and species, and the unexplained variance (residual variance). The residual variance comprises analytical error and various types of intraspecific variation including micro-site and within-canopy variation. Where necessary, variables were log transformed to meet assumptions of ANOVA (see Table 1 for details). In addition, permutation-based multivariate analysis of variance (PERMANOVA; Anderson 2001) was applied to the matrix of dissimilarity among traits to evaluate the importance of soil type, species identity and the interaction soil-species as a source of variation in the 24 traits simultaneously. The non-parametric permutation-based analysis of variance (PERMANOVA) was then performed on the resulting distances (10000 permutations). An alpha level of 0.05 was used for all significance tests, and no effort was made to test for or address non-normal data distributions. The PERMANOVA used distance matrices calculated using the adonis function in the vegan package of R.

Leaf traits were grouped using principal component analysis (PCA) using Simca-P (2016) software (Umetrics Ltd, Sweden). The principal components for the variables were obtained by the correlation matrix modelling in lieu of covariance matrix modelling. We used the unit variance scaling (van den Berg, Hoefsloot, Westerhuis, Smilde, & van der Werf, 2006) to avoid the effects of variables with high variance. The PCA was used to obtain score scatter, loadings, as well as R2 and Q2 overview plots to evaluate whether traits clustered into functional groups. R2 values denote how well a trait can be explained in the model and Q2 denote how well a trait can be predicted from the dataset."

Referee comment: Line 21 – "Foliar traits were predicted from spectral reflectance using partial least square regression, and so inter- and intra-specific variation." – Presumed typo – rewrite.

Author response: We have changed the text to: Foliar traits were predicted from spectral reflectance data using partial least square regression.

Referee comment: Line 22 – "Most leaf traits varied greatly among species" – a) replace 'among species' with either within or between species (presumed between?) b) Also this sentence is confusing – suggests that there was simply a wide variation in leaf trait measurements - slightly random to mention in abstract. Actual meaning I think is something along these lines "Inter-specific variation was the largest contributor to trait variation".

Author response: We have altered the sentence to ".  Analysis of variance showed that inter-specific differences in traits were generally much stronger than phenotypic differences related to soil type, accounting for 25% versus 5% of total trait variation, respectively." in the abstract and further explained in the results.

Referee comment: Line 23 – "Macronutrient concentrations were greater on alluvial than chalk soils while micronutrient concentration showed the opposite trend" – Foliar macronutrient concentrations or soil macronutrient concentrations? (presumed the former?). Also, slightly odd sentence – what's the significance? Maybe meant to say something along these lines? - "However, foliar macro- and micronutrient concentrations were found to be more strongly influenced by soil type".

Author response: We have changed the text to: "foliar concentrations of rock-derived nutrients did vary: P and K concentration were lower on chalk than alluvial soils, while Ca, Mg, B, Mn and Zn concentrations were all higher, consistent with the findings of previous ecological studies."

Referee comment: Line 24 – "Si predictions using spectroscopy appear to be promising" – what's so special about Si predictions?! Why do they get singled out?

Author response: It was the first time Si was reported as a trait able to be predicted using spectroscopy in forests. But we agree that it should not be singled out as Si is not the main focus for the paper. We have rephrased the sentences that were mentioning Si as a very important finding and we have reduced the amount of text on Si in the discussion.

Referee comment: Line 28 – "However, it [field spectroscopy] was unable to detect subtle within species variation of traits associated with soil type" – repetition of line 25? ("Field spectroscopy. . ..was less effective at detecting subtle variation of rock-derived nutrients between soil types"). Combine sentences to keep abstract concise?

Author response: We have rephrased it and considerably expanded the text on the lack of detection of subtle variation of rock-derived nutrients due to soil variation as follows: "Some of most accurately predicted traits have no absorption features in the visible-to-near-infrared, but were instead estimated indirectly via constellation effects. Rock-derived nutrients lack absorption features in visible to shortwave-infrared region of the electromagnetic spectrum so cannot be measured directly by spectroscopy. They can, nevertheless, be estimated indirectly by virtue of the fact that element concentrations co-vary with organic molecules that do have strong absorption features ("constellation effects", see above). This paper identifies a problem with this approach: there were strong differences in rock-derived mineral nutrients between soil types, but we could not measure these because the concentrations of defence and structural traits were barely affected by soil type. We have shown many similarities between our study and those in tropical forests, demonstrating that this problem is likely to be widespread."

Introduction

Referee comment: Line 64 – "along environmental change". Typo. Suggested "along environmental gradients"?

Author response: We have changed the text to "In response to environmental change".

Referee comment: Line 71 – "However, spectral and chemical properties may be uncoupled if intraspecific variation in foliar traits is high and/or phenotypic plasticity exceeds

phylogenetic patterns among leaf properties". Disagree. Spectral and chemical relationships would still hold, it would just be harder to identify species type based on their reflectance signatures.

Author response: We agree with you. We found references supporting your statement: "Wu et al., (2016) found that leaf traits and spectra vary systematically and predictably with leaf age between forest sites on contrasting soil types thousands of kilometers apart." Madritch et al. (2014) also demonstrated the high correlation between spectral properties, traits and soil very well.

Structural differences (i.e., leaf thickness, number of air water interfaces, cuticle thickness, and pubescence) between leaves may have significant effects on the relationship between leaf reflectance and traits, and can complicate interpretation of data (Sims & Gamon, 2002; Wu et al., 2016). The ability of spectroscopy to measure phenotypic variation in multiples traits between soil types, particularly when some of those traits are indirectly determined through constellation effects, has not been critically evaluated.

Madritch, M.D., Kingdon, C.C., Singh, A., Mock, K.E., Lindroth, R.L. and Townsend, P.A., 2014. Imaging spectroscopy links aspen genotype with below-ground processes at landscape scales. *Philosophical Transactions of the Royal Society of London B: Biological Sciences*, *369*(1643), p.20130194.

Wu, J., Chavana-Bryant, C., Prohaska, N., Serbin, S.P., Guan, K., Albert, L.P., Yang, X., Leeuwen, W.J., Garnello, A.J., Martins, G. and Malhi, Y., 2016. Convergence in relationships between leaf traits, spectra and age across diverse canopy environments and two contrasting tropical forests. *New Phytologist*.

Referee comment:  Line 73 – "Martin and Aber (1996) demonstrated that equations for estimating leaf properties from one site were unable to predict leaf properties for other sites, due to variability in the magnitudes of foliar traits levels between data sets and environmental influences". Very old reference and what about all the evidence to the contrary (e.g. all of Asner's work) ???

Author response: As per previous comment, we have deleted that reference. We also agree that we can find high correlation between spectral properties, traits and soil (see previous comment).

Referee comment: Line 75 – "To our knowledge, the link between foliar traits and spectral properties of trees has not been broadly demonstrated for temperate forests" – query this statement. The remote sensing of foliar traits began in temperate forests.

Author response: We are no longer making this statement, which can be proved wrong indeed. "There is currently great interest in hyperspectroscopy in vegetation science, particularly because improved airborne sensors and faster computing make it possible to map functional traits from the air (Asner and Martin, 2016; Jetz et al., 2016; Asner et al., 2017)". We have slightly changed the focus of the paper. "The ability of spectroscopy to measure phenotypic variation in multiples traits between soil types, particularly when some of those traits are indirectly determined through constellation effects, has not been critically evaluated."

Referee comment: Line 86 – "what is the relative contribution of soil type and species to leaf trait variation?". Missed word? "what is the relative contribution of soil type and species type to leaf trait variation".

Author response: We have included the word species identity.

Referee comment: Line 88 – "does the importance of the three functional groups change due to soil or more due to species variation?" – awkward phrasing. Rephrase.

Author response: We have rephrased all the sentences involving functional groups.

Material and Methods

Referee comment: Line 102: "Leaves of 66 trees of six species were collected from the two contrasting soil types. The six species were in common to both sites". Suggested "Across both sites, leaves were collected from 66 trees, representing six species. The six species common to both sites were:"

Author response: Many thanks. We have made this amendment.

Referee comment: Line 103: "Acer campestre L. (Field Maple)" – what does the L stand for?

Author response: L. is the authority - the person who named the species formally. In this case Linnaeus, who back in the 1700s invented the Latin binomial system for naming species that is still used today. Some biology journals insist on including these. We have removed them from this paper

Referee comment: Line 105: "Two fully sunlit branches were selected, were cut and placed on ice in a cool box, and transported to a lab for processing within 2 hours (and often within 30 minutes)".

Author response: We have changed the text to: Two fully sunlit branches were selected, cut and placed in a cool box, and subsequently transported to a laboratory for processing within two hours.

Referee comment: Line 149: "2.4 Statistical analyses". Needs to be split up into each statistical analysis performed and titled accordingly.

Author response: This has been carried out as requested.

Referee comment: Line 156: "Where necessary, variables were log transformed to meet assumptions of ANOVA".

Author response: Table 1 has additional information concerning which variables were log transformed and how they can be found.

Referee comment: Line 168: PLSR section – no mention of using 70% to calibrate and 30% to test but Cal and Val appear on Table 3. No mention of how the data for Figure 3 and 4 is acquired!!!

Author response: We added the following sentences to the text to make it clearer: "We adopted a leave-one-out cross-validation for each PLSR model. Model accuracy and precision were expressed by the coefficient of determination ($R^2$) and root mean square error (RMSE). We also standardised RMSE to the percentage of the response range (RMSE%) by dividing each RMSE by the maximum and minimum values of each leaf trait,

as in Feilhauer et al. (2010). RMSE and $R^2$ were acquired during both model calibration and after model validation.”

Regarding Figure 3: We have rephrased the text to: “To evaluate the correlation among traits, Spearman rank correlation coefficient was calculated between all trait pairs and the variables were ordered in the figure by hierarchical clustering.”

Regarding Figure 4: Picture 4 no longer exists as previously explained.

Results

Referee comment:   Line 204 – “Species exerted little or no influence on pigment concentrations” – Refer to species in this context (and throughout paper) as ‘species type’?

Author response: Yes, it does. We have changed this throughout the document to species identity.

Referee comment:   Line 241: “Ability to predict leaf traits from hyperspectral reflectance varied greatly among the 24 traits fitted using the 6 species (Table 3)”. “fitted using the 6 species” - confusing. Rephrase.

Author response: We have removed the “fitted using the 6 species” as it did not make any sense.

Referee comment:  Line 243: “PLSR modelling for LMA, water, Si, phenolics, carotenoids, K, B, efficiency of PSII, N, chlorophyll a and chlorophyll b were in descending order the best performing in terms of”

Author response: Thank you. We have corrected the text.

Referee comment:  Line 248- “higher goodness-of-fit” – use a different term? E.g. stronger relationships/correlations etc.

Author response: OK – we have changed the text to higher strength of relationship.

Referee comment: Line 256: "There were strong correlations among some of the leaf properties (Fig. 3) that can be potentially leveraging the estimation of other leaf traits from the use of PLSR". Interesting. Explain further?

Author response: We added the following sentences in the results: "Some leaf traits which appeared to be predicted accurately by PLSR do not have absorbance features in the 400-2500 nm range, and were instead predicted because of their close association with leaf traits that do have absorbance features in that range (see correlations in Fig. 4). For instance, Si and B do not have absorption features in the 400-2500 nm range, but their concentrations are highly correlated to hemicellulose, cellulose and lignin concentrations, and these organic polymers do have strong absorbance features in the SWIR region. Likewise, K do not have absorption features in the 400-2500 nm range, but K concentration is highly correlated to leaf water content, soluble carbon, lignin, hemicellulose and cellulose, all of which have absorbance features in the region. The importance of these "constellation effects" (sensu Chadwick and Asner 2016) becomes apparent when we examine the partitioning of variance of PLSR-predicted trait values: several rock-derived nutrients vary significantly with soil type when measured in leaves (Fig. 1) but little of that variation is successfully modelled by PLSR (Fig. 5). The explanation for this failure to model soil-related variation correctly is that concentrations of their associated traits remain invariant of soil type (Table 1). The use of PLSR also considerably under-predicted the importance of soil (~ 37 %) on the δ15N variation, presumably for similar reasons. Some species-soil interaction effects were detected by PLSR modelling, except for traits that showed strong interaction (Mn, P and δ13C). PLSR models were better able to detect intra-specific variation in foliar N concentrations, because much of the nitrogen is contained in proteins, which have strong absorbance features."

Referee comment: Line 257: "The correlation graphic also shows the similarity among variables through cluster analysis". Explain. Cluster analysis was not been mentioned in the Materials and Methods. Explain how this was achieved, why it was done and expand on results.

Author response: We have made it clearer in the Material and methods as follows: "To evaluate the correlation among traits, Spearman rank correlation coefficient was calculated between all trait pairs and the variables were ordered in the figure by hierarchical clustering."

Discussion

Referee comment: Line 271: "Some leaf traits were strongly influenced by both species and soil type, while others were hardly affected by soil and only varied with species". Vague. Make more specific.

Author response: We added many references on the P supply implications in chalk soils as follows: "Compared with trees growing on deep alluvium, trees on thin chalk soils had low concentrations of N, P and K macronutrients in their leaves, but high concentrations of several micronutrients. Similar findings have been reported for herbaceous species growing on chalk (Hillier, Walton, & Wells, 1990)." And "Compared with trees growing on deep alluvium, trees on thin chalk soils had low concentrations of N, P and K macronutrients in their leaves, but high concentrations of several micronutrients. Similar findings have been reported for herbaceous species growing on chalk (Hillier et al., 1990)."

Referee comment: Line 305: "water" – change to 'leaf water content'.

Author response: Thank you. We have done all the corrections ad changed the term "water" to "leaf water content" throughout the text.

Referee comment: Line 321: "but their study sampled only from fully sunlit leaves". Suggested - "Similarly, their study sampled only from fully sunlit leaves".

Author response: Thank you. Alteration made.

Referee comment: Line 325: "The investment in light capture had high intra-specific variation, and neither species nor soil accounted for variation in [these] foliar properties". Missing word.

Author response: We have restructured the sentence.

Referee comment: Line 327: "separating out some species". Confusing. Rephrase?

Author response: We have improved the discussion and included the following sentence that had the same meaning: "The investment in structure and defence-related traits were little influenced by soil type and was mainly determined by species identity." "The traits most influenced by species (in descending order) were Si, leaf water content, B, soluble C, N, LMA, K, cellulose, lignin, hemicellulose, magnesium, Zn, phenolics and Fe."

Referee comment: Line 327: "Investment in traits related to defence and leaf structure is species-mediated, and may be separated into two defensive strategies". State the two defensive strategies?

Author response: We improved the discussion regarding the functional grouping as follows: "Species had a greater influence on trait values than soils for all traits except P, and PCA analyses demonstrated that species with traits associated with fast growth had low concentration of traits associated with defence and structure (see Coley 1983; 1987; Fine et al. 2006). Traits favouring high photosynthetic rate and growth are usually considered advantageous in rich-resource soil environments, while traits favouring resource conservation are considered advantageous in low-resource environments (Aerts & Chapin, 1999; Westoby, Falster, Moles, Vesk, & Wright, 2002), but in this study the species were generalists growing on both soil types. "

Referee comment: Line 342: "Doing so revealed that. . .". Awkward. Rephrase.

Author response: This sentence does not exist anymore after the changes throughout the discussion.

Referee comment: Line 351: "Although chlorophylls also contain nitrogen, the spectra of chlorophylls differ greatly from proteins because of their dissimilar chemical structures, showing strong absorption due to C-H bonds in the phytol tail of the molecule (Katz et al., 1966), also confirmed in this work when visualizing the regions of importance for predictions." Require a full stop after (Katz et al. 1996) and develop last sentence ("also confirmed in this work when visualizing the regions of importance for predictions").

Author response: We have refined the text to "Although chlorophylls also contain nitrogen, the spectra of chlorophylls differ greatly from proteins because of their dissimilar chemical structures, showing strong absorption due to C-H bonds in the phytol tail of the molecule

(Katz, Dougherty, & Boucher, 1966). That can be confirmed in this work as the visible region of the spectrum showed the best predictions of pigments."

Referee comment: Line 360: "On the other hand, the use of spectroscopy on fresh leaves is particularly better for LMA predictions"

Author response: We have edited the paragraph, which included that specific sentence to "Leaf mass per unit area (LMA) is consistently among the more accurately predicted traits using spectroscopy (ASNER & Martin, 2008; Chavana-Bryant et al., 2016; Serbin, Singh, McNeil, Kingdon, & Townsend, 2014),  but is measured indirectly via its close coupling with water content and leaf structural traits (Asner et al. 2011b)."

Referee comment:  Line 365: "The use of spectroscopy for Si predictions on fresh leaves appears to be promising considering our accurate results". Maybe, but why are Si predictions so important? What ecological function does Si perform?!

Author response: We have reduced the text on Si to avoid singling it out, as it is not a specific question on this paper.

Referee comment:  Line 339: 4.4 Predictions of foliar traits using spectroscopy – this section maybe a bit long? Could condense? Says some interesting things but I'm not sure they're all relevant to the paper.

Author response: We have condensed and restructured the entire section named "Measuring interspecific variation in leaf traits with field spectroscopy". We have also discussed the Difficulties in measuring phenotypic variation by field spectroscopy and its implications for mapping functional traits.

Referee comment:  Line 384: Consideration on the use of spectroscopy to quantify patterns of foliar traits. Typo - Consideration of the use of spectroscopy to quantify patterns of foliar traits.

Author response: Thanks. The correction was made.

Referee comment:  Line 385. "The range of variation within species for most predicted traits tend to be smaller with the use of PLSR on reflectance". Very confusing. Rephrase.

Referee comment: Line 399: "This study particularly provides findings for a large range of traits that indicate that the use of spectroscopy may be useful to quantify structural traits but can be misleading to measure the environmental filtering on traits that are indirectly predicted, such as macro- and micronutrients". I might agree if I understood Figure 4 but, as I don't, I query this statement.

Author response: The updated Figure 4 is Figure 5. However, we have changed to picture to one that can be easily interpreted. We have restructured the text to make the Figure 4 clearer.

[Figure]

Figure 5. Partitioning of variance of foliar traits between species, soil, species-soil interaction and residual components for six generalist species found on both chalk and alluvial soils from predicted data. Residual variation arises from within-site intraspecific variation, microsite variability, canopy selection and measurement error variance. Predicted data were obtained from partial least square regression (PLSR).

Referee comment:   Line 401: "While remote sensing is not a direct replacement of field sampling, the ability of remote sensing platforms to assess biological phenomena at large spatial scales is unparalleled". Slightly random – doesn't follow from previous statement/results section.

Author response: We agree that it completely disagree with previous statements and results. We are no longer including this statement on the paper.

Conclusion

Referee comment:   Line 407: "rock-derived nutrients are strongly influenced by the soil characteristics". Need to tone down or change previous sentence, otherwise statements are contradictory.

Referee comment:  Line 409: "This study also demonstrates the potential for estimating foliar traits by field spectroscopy and its promising use to predict Si". a) "demonstrates the potential" –this has already been done many times. Maybe something more along the lines of "agrees with the existing literature in demonstrating the potential. . ." b) "its promising use to predict Si". Once again – what is so important about Si?!?!?!

Author response: We no longer have the conclusion but we decided to expand the discussion on the Difficulties in measuring phenotypic variation by field spectroscopy and its implications for mapping functional traits

Figures

Referee comment:   Line 661: "Red and black circles mean negative and positive correlations". Which way round?

Referee comment:  Line 668: "The greyness and size of each dot reflects the goodness-of-fit of the PLSR for each foliar trait, with darker and bigger points representing the most

accurate PLSR predictions. goodness-of-fit". Give statistical boundaries for how dots were sorted into each size/shape category.

Author response: We have the changed the figure to:

[Figure]

Figure 5. Partitioning of variance of foliar traits between species, soil, species-soil interaction and residual components for six generalist species found on both chalk and alluvial soils from predicted data. Residual variation arises from within-site intraspecific variation, micro-site variability, canopy selection and measurement error variance. Predicted data were obtained from partial least square regression (PLSR).

Referee comment: Perhaps add the word "…respectively" at the end to clarify which is which?

Author response: Thank you. The correction was made.

Referee comment: Line 675: Table 1. CV needs to be represented as %CV, as stated in the heading.

Author response: Thank you. It is corrected now.

**Response to Anonymous Referee #2's comments**

Referee comment: The article "Leaf trait variation and field spectroscopy of generalist tree species on contrasting soil types" by Nunes and co-authors analyzed field spectroscopy data collected on different European tree species on contrasting soil types. The authors worked with 24 leaf traits and explored the following questions: What contribution do soil type and species identity make to trait variation? When traits are clustered into three functional groups (light capture and growth, leaf structure and defence, as well as rock-derived nutrients), are some groups more affected by soil than others? What traits can be estimated precisely using field spectroscopy? Can leaf spectra be used to detect inter-soil as well as inter-specific variation in traits? The authors found that most leaf traits varied greatly among species. The effects of soil type were generally weak by comparison.

Specific Comments:

Referee comment: Line 28 variation in foliar traits and Si predictions using spectroscopy appear to be promising. Not clear what Si means at this stage, it becomes clear later. But in general all the discussion on Si is poor

Author response: Firstly, we spelled out Si and all the nutrients that were presented on the paper as an acronym. We previously singled out the performance of Si as a promising result but its performance should not be the main focus of the manuscript. We have reduced the discussion on Si and broadened the review out to include other traits.

Referee comment: Line 162 We recognize that grouping leaf properties into functional classes can be controversial, given that a single leaf property can contribute to. This is

particularly true for P, this assumption has to be justified as foliar P can be easily considered a trait associated to growth.

Author response: An increasing number of leaf traits are measured routinely in plant communities and global tradeoffs among these traits are often interpreted in terms of life history of different species (Adler et al., 2014; Aubin, Ouellette, Legendre, Messier, & Bouchard, 2009; Fry, Power, & Manning, 2014; Pillar, Sosinski, & Lepš, 2003). In this study we measured 24 traits which we organise into three functional groups (Gregory P. Asner et al., 2015; Gregory Pa Asner, 2014). We recognise that leaf traits can contribute to more than one class (e.g. LMA is related to growth but also to defence, P is a rock-derived nutrient also associated with growth). Leaf traits were grouped using principal component analysis (PCA) using Simca-P (2016) software (Umetrics Ltd, Sweden). The principal components for the variables were obtained by the correlation matrix modelling in lieu of covariance matrix modelling. We used the unit variance scaling (van den Berg et al., 2006) to avoid the effects of variables with high variance. The PCA was used to obtain score scatter, loadings, as well as R2 and Q2 overview plots to evaluate whether traits clustered into functional groups. R2 values denote how well a trait can be explained in the model and Q2 denote how well a trait can be predicted from the dataset.

Results Section Spectroscopy of leaf properties

Referee comment: The results of PLSR are on one hand encouraging because the portion of spectra selected for specific traits are in line with what expected from the literature. Some examples from the article: 1) higher goodness-of-fit were obtained for K, Ca and P in the SWIR regions. 2) Pigments were the only traits that predictions were more accurate when using the visible region (400 – 700 nm)

Author response: Many thanks. We thought this encouraging too.

Referee comment: I think would be useful to have more discussion on what is known and what is new compared for instance to the review from Homolova et al., which discuss many of the traits mentioned by the authors and how these traits can be predicted from remote sensing data. What do we learn from these results? I think the authors should make an effort to improve this aspect because can be quite relevant considering the great dataset they have. For example a figure with a reflectance spectra with an indication of the regions relevant to estimate other the traits indicated might be useful for the reader.

Author response: I agree that it would be interesting to have a figure with an average reflectance spectrum indicating the relevant regions for each trait, as per Figure 1. We included the Coefficient of variation  (%) and the average reflectance with the regions

partitioning indicating which part of the spectrum is more suitable for each trait. There are amendments in the Material and Methods, as well as Results sections on the graphic.

[Figure]

Figure 3. Spectral reflectance and coefficient of variation (% CV) of reflectance of six generalists species for alluvial and chalk soils. The spectral regions for each trait were selected based on the model that minimised RMSE.

Referee comment:  Line 267 The species x soil interaction effects were detected by PLSR modelling, except for traits that showed strong interaction (Mn, P and δ 13C). This should be better discussed

Author response: We have improved considerably the discussion on the ability of field spectroscopy to predict trait variation. The following paragraph was added to the results: "Some leaf traits which appeared to be predicted accurately by PLSR do not have absorbance features in the 400-2500 nm range, and were instead predicted because of their close association with leaf traits that do have absorbance features in that range (see correlations in Fig. 4).  For instance, Si and B do not have absorption features in the 400-2500 nm range, but their concentrations are highly correlated to hemicellulose, cellulose and lignin concentrations, and these organic polymers do have strong absorbance features in the SWIR region. Likewise, K do not have absorption features in the 400-2500 nm range, but K concentration is highly correlated to leaf water content, soluble carbon, lignin, hemicellulose

and cellulose, all of which have absorbance features in the region. The importance of these "constellation effects" (sensu Chadwick and Asner 2016) becomes apparent when we examine the partitioning of variance of PLSR-predicted trait values: several rock-derived nutrients vary significantly with soil type when measured in leaves (Fig. 1) but little of that variation is successfully modelled by PLSR (Fig. 5). The explanation for this failure to model soil-related variation correctly is that concentrations of their associated traits remain invariant of soil type (Table 1). The use of PLSR also considerably under-predicted the importance of soil (~ 37 %) on the $\delta15N$ variation, presumably for similar reasons.  Some species-soil interaction effects were detected by PLSR modelling, except for traits that showed strong interaction (Mn, P and $\delta13C$).  PLSR models were better able to detect intra-specific variation in foliar N concentrations, because much of the nitrogen is contained in proteins, which have strong absorbance features. ".

And this to the discussion: "Rock-derived nutrients lack absorption features in visible to shortwave-infrared region of the electromagnetic spectrum so cannot be measured directly by spectroscopy.  They can, nevertheless, be estimated indirectly by virtue of the fact that element concentrations co-vary with organic molecules that do have strong absorption features ("constellation effects", see above). This paper identifies a problem with this approach: there were strong differences in rock-derived mineral nutrients between soil types, but we could not measure these because the concentrations of defence and structural traits were barely affected by soil type. We have shown many similarities between our study and those in tropical forests, demonstrating that this problem is likely to be widespread.

What are the implications of the constellation-effect problem for mapping functional traits using imaging spectroscopy? Ever larger areas of earth are being mapped with airborne spectrometers (e.g. Asner et al. 2017) and the anticipated launch of satellite-borne sensors (e.g. EnMAP; DLR 2015; Guanter et al. 2015) will soon enable vegetation and ecosystem function to be characterised at a global scale. The effectiveness of indirect prediction of traits using constellation-effect will depend critically on whether soils act as a strong filter on tree species within a particular region. In the Amazonian lowlands, Asner et al. (2015) found that variation in soil P was mirrored by changes in species composition, and that P variation among species was correlated with changes in structural and defence compounds: in this instance, indirect estimation should be effective (e.g. Dana Chadwick & Asner 2016).  On the other hand, in low-diversity temperate forests, a single tree species is often found to span many different soil types and show substantial phenotypic plasticity in some traits (Oleksyn, Reich, Zytkowiak, Karolewski, & Tjoelker, 2002; Turnbull et al., 2016). The six species growing on both chalk and alluvial soils in this study are a case in point.  In these low diversity systems, it will be much more difficult to map variation using constellation

effects, for the reasons explained above. Our study confirms the power of spectroscopy for predicting biochemical and structural plant traits, but we urges caution in interpreting results when species range across contrasting soil types. "

Referee comment: Line 279 Our findings that trees growing on the chalk soils had relatively low concentrations of N, P and K in their leaves, and relatively high concentrations of Ca, Mg, B, Mn, Si and Zn, is consistent with previous analyses of mineral nutrition in calcareous soils. Please add a reference here

Author response: Compared with trees growing on deep alluvium, trees on thin chalk soils had low concentrations of N, P and K macronutrients in their leaves, but high concentrations of several micronutrients. Similar findings have been reported for herbaceous species growing on chalk (Hillier et al., 1990).

Referee comment: The discovery that structural and defensive traits do not vary with soil is consistent with a previous study in New Zealand's lowland temperate rain forests (Wright et al., 2010). That study compared traits of trees growing on phosphorus rich alluvium versus phosphorus-depleted marine terraces. Foliar phosphorus concentrations of species were halved on the marine terraces, but there was no detectable variation in structural traits, phenolic or tannin concentrations. I would add more discussion at line 298. At the moment is more a description of results. Please specify at the beginning which traits are you talking about and why they do not change between poor and rich soils:

Author response: We have added more references and made the sentences clearer: "Importantly for our later discussion on indirect estimation of traits by spectroscopy, species did not vary between soil types in their structural and defensive traits (i.e. LMA, lignin, phenolics) despite these differences in rock-derived nutrients. A similar lack of phenotypic change has been found in New Zealand rainforest trees growing on alluvium versus phosphorus-depleted marine terraces (Wright et al., 2010) and in several other studies (Boege & Dirzo, 2004; Fine et al., 2006; Koricheva, Larsson, Haukioja, & Keinanen, 1998)."

Referee comment: "Water" was defined as trait. Please define exactly what do you mean with water and how this was computed also here

Author response: We included the following sentence on the paper: "Leaf water content was computed as the ratio between the quantity of water (fresh weight – dry weight) and the fresh weight." We also used the term leaf water content throughout the paper.

Referee comment: Line 304: Species had a greater influence on trait values than soils for all traits, except P. This makes completely sense to me because the content of P in leaves

should be more related to the P available in the soil for the plants and not too much to the species. But again I found the discussion poor. There is a lot of literature about the leaf stoichiometry and P stoichiometry and the relationship with physical and chemical properties of the soil.

Author response: We agree that some discussion on P was missing out and we expanded the soil and species effects on traits considering that P supply limitation: "Compared with trees growing on deep alluvium, trees on thin chalk soils had low concentrations of N, P and K macronutrients in their leaves, but high concentrations of several micronutrients. Similar findings have been reported for herbaceous species growing on chalk (Hillier et al., 1990). Phosphorus and several micronutrients form low-solubility compounds in alkaline soils and become less available for plant uptake (Marschner, 1995; Misra & Tyler, 2000; Sardans & Peñuelas, 2004; Tyler, 2002), while the low N concentrations may reflect stoichiometric constraints (Niklas, Owens, Reich, & Cobb, 2005). The lower efficiency of PSII in the chalk soil is likely to be consequence of phosphorus deficiency (Santos et al. 2006). Importantly for our later discussion on indirect estimation of traits by spectroscopy, species did not vary between soil types in their structural and defensive traits (i.e. LMA, lignin, phenolics) despite these differences in rock-derived nutrients. A similar lack of phenotypic change has been found in New Zealand rainforest trees growing on alluvium versus phosphorus-depleted marine terraces (Wright et al., 2010) and in several other studies (Boege & Dirzo, 2004; Fine et al., 2006; Koricheva et al., 1998).

Species had a greater influence on trait values than soils for all traits except P, and PCA analyses demonstrated that species with traits associated with fast growth had low concentration of traits associated with defence and structure (see Coley 1983; 1987; Fine et al. 2006). Traits favouring high photosynthetic rate and growth are usually considered advantageous in rich-resource soil environments, while traits favouring resource conservation are considered advantageous in low-resource environments (Aerts & Chapin, 1999; Westoby et al., 2002), but in this study the species were generalists growing on both soil types. The traits most influenced by species (in descending order) were Si, leaf water content, B, soluble C, N, LMA, K, cellulose, lignin, hemicellulose, magnesium, Zn, phenolics and Fe. It is interesting to note that two trace elements were near the top of this list; it is likely that strong differences in B and Si concentrations between species reflect differences in ion channel activity in roots (Ma & Yamaji, 2006). Previous studies have also shown Si to be under strong phylogenetic control, and to be little affected by environmental conditions (Hodson, White, Mead, & Broadley, 2005). We also found Si and B concentrations to be positively correlated, which might ameliorate the effects on B toxicity as Si can increase B tolerance of plants (Gunes, Inal, Bagci, Coban, & Sahin, 2007). High Zn organization at the

species level corroborates earlier analysis that show more than 70% of Zn variation occurs within family and substantial differences exist between and within species (Broadley, White, Hammond, Zelko, & Lux, 2007).

The patterns revealed by our variance partitioning analysis of six temperate species (Fig. 1) bear surprising similarities to those emerging from an analysis of 3246 species from nine tropical regions (Fig. 5 of Asner & Martin 2016a). The tropical analyses included a "site" term which captured variation due to soil and geology, among other factors. They, like us, found that taxonomic identity explained far more variation than site for most traits. They, like us, found foliar concentrations of P and other rock-derived minerals varied strongly with site, while nitrogen concentrations varied little. They, like us, found that soluble carbon, structural and defensive traits hardly varied between sites. And they, like us, observed that pigments (in their case just chlorophyll) was the least predictable of traits, probably because photosynthesis is rapidly up- and down-regulated in response to light environment among other factors (Gregory P Asner & Martin, 2011). Similarly, δ13C is known to vary strongly with light condition and with relative humidity (Buchmann, Kao, & Ehleringer, 1997; Yan et al., 2012) which may explain why species and soil explained little of its variance in our study. These parallels between tropical and temperate systems suggest broad similarities in plant responses to soil across different regions that differ greatly in temperature. "

Referee comment: Also with the database the authors have they can also explore how the reflectance is related to ratio such as C/P N/P or C/N ratios.

Author response: Unfortunately, P is not well predicted; the few studies spectroscopy studies available differ in the spectral bands they chose to model P (Homolová, Malenovský, Clevers, García-Santos, & Schaepman, 2013). RNA and DNA absorb in the ultraviolet (e.g. Tataurov et al. 2008) and phosphates in the longwave infrared, but there are no pronounced absorption features in the VSWIR region (Homolová et al., 2013) and covariance with other traits is weak so constellation effects are unreliable. Rock-derived nutrients lack absorption features in visible to shortwave-infrared region of the electromagnetic spectrum so cannot be measured directly by spectroscopy. They can, nevertheless, be estimated indirectly by virtue of the fact that element concentrations co-vary with organic molecules that do have strong absorption features ("constellation effects"). Because of the confounding factors revolving around rock-derived nutrients predictions, we decided not to include ratios that would not be directly detected.

Referee comment: Line 350 The region of importance with correlated wavelengths with nitrogen varies between 1192 nm in deciduous forest (Bolster et al., 1996) to 2490 for forage matter (Marten et al., 1983), which results directly from nitrogen in the molecular structure.

Please also cite other recent papers showing similar results with spectrometers similar to the one used in this study (e.g. Homolova et al., 2013).

Author response: Thank you for the suggestion. We have included it: "According to Kumar et al. (2001), three main protein absorption features report as important for N estimation are located around 1680 nm, 2050 nm and 2170 nm."

Referee comment: Line 353 Although chlorophylls also contain nitrogen, the spectra of chlorophylls differ greatly from proteins because of their dissimilar chemical structures, showing strong absorption due to C-H bonds in the phytol tail of the molecule (Katz et al., 1966), Here if I understand correctly the authors they want to make the point that Chl and N are estimated with different regions of the spectrum despite N is one component of Chl and should covary. If my interpretation is correct I suggest another line of argumentation: Nitrogen Chl are contained in the green vegetation and N content and Chl are correlated (see Houborg et al., 2013). However, in dry leaves there is only N and not Chl. And therefore we cannot expect that the PLSR select similar regions for Chl and N.

Author response: We have added the following sentences to the text: "The region of importance with correlated wavelengths with nitrogen varies between 1192 nm in deciduous forest (Bolster et al., 1996) to 2490 for forage matter (Marten et al., 1983), which results directly from nitrogen in the molecular structure. According to Kumar et al. (2001), three main protein absorption features reported as important for N estimation are located around 1680 nm, 2050 nm and 2170 nm. Although chlorophylls also contain nitrogen, the spectra of chlorophylls differ greatly from proteins because of their dissimilar chemical structures, showing strong absorption due to C-H bonds in the phytol tail of the molecule (Katz et al., 1966). That can be confirmed in this work as the visible region of the spectrum showed the best predictions of pigments."

Chl and N were not correlated in our study and the spectral measurements were done on fresh leaves. The main reason for PLSR to select different regions was that N is correlated to the proteins and Chl (even though they contain nitrogen) to the phytoil tails.

[revised manuscript text omitted]

---

## Author Response (AR2)

Dear Dr Bahn,

Thank you for your email regarding the manuscript *On the challenges of using field spectroscopy to measure the impact of soil type on leaf traits.*

We were pleased that the revised paper has introduced many of the comments as mentioned by referee #2. We are grateful to the referee for suggesting a relevant paper that we then used to improve the discussion. We also have placed the main conclusions in a separate section as you suggested. The marked-up version has been placed below the point-by-point reply to the comments and changes were highlighted.

We thank the referee again for the insightful suggestion, and hope that our revisions will be satisfactory.

Yours sincerely

Matheus Henrique Nunes and David Coomes.

**Response to Anonymous Referee #2's comments**

**General Comments :**

Referee comment: The article is substantially improved after the revision. The authors revised carefully the manuscript and introduced many of the comments ad revisions suggested.

Author response: We thank the referee for these positive comments.

Referee comment: I suggest to the authors to carefully check in Fig 3 of the response to the reviewer the spectral range that they indicate as relevant to detect the efficiency of the PSII. Please check Porcar Castell et al., 2014, which discuss these aspects in details.

Albert Porcar-Castell, Esa Tyystjärvi, Jon Atherton, Christiaan van der Tol, Jaume Flexas, Erhard E. Pfündel, Jose Moreno, Christian Frankenberg and Joseph A. Berry. Linking chlorophyll a fluorescence to photosynthesis for remote sensing applications: mechanisms and challenges. Journal of Experimental Botany, Vol. 65, No. 15, pp. 4065–4095, 2014 doi:10.1093/jxb/eru191

Author response: Many thanks for the suggestion. Porcar-Castel et al.'s paper was indeed very relevant to the understanding of the relationship between efficiency of PSII and absorption features. We have added some information in the discussion as follows:

"In this study, pigments were found to influence the visible region of the spectrum while PSII-efficiency was predicted from features across the VSWIR range. The spectra of chlorophylls are distinct from those of proteins because C-H bonds in their phytols tails create a strong absorption feature not found in proteins (Katz et al. 1966). However, pigments are tightly bound by proteins to form photosynthetic antenna complexes that capture light energy and transfer it to the PSI and PSII reaction centres (Liu et al., 2004). The vibration of the bonds in the pigment–protein complex adds additional absorption features to the spectra of pigments and may help explain why so many bands were involved in PSII-efficiency prediction (Porcar-Castell et al., 2014)."

[revised manuscript text omitted]
,In this study, pigments were found to influence the visible region of the spectrum while PSII-efficiency was predicted from features across the VSWIR range.  The spectra of chlorophylls differ greatlyare distinct from those of proteins because ofC-H bonds in their dissimilar chemical structures, showingphytols tails create a strong absorption due to C-H bonds in the phytol tail of the moleculefeature not found in proteins (Katz et al. 1966). That can be confirmed in this work as the visible region of the spectrum showed the best predictions of pigments. The 1500-1900 nm region was alsoHowever, pigments are tightly bound by proteins to form photosynthetic antenna complexes that capture light energy and transfer it to the PSI and PSII reaction centres (Liu et al., 2004). The vibration of the bonds in the pigment–protein complex adds additional absorption features to the spectra of pigments and may help explain why so many bands were involved in PSII-efficiency prediction (Porcar-Castell et al., 2014). The 1500-1900 nm region was important for phenolic compounds prediction, which includes the 1660 nm feature across

a variety of species and phenolic compounds (Windham et al. 1988; Kokaly & Skidmore 2015). The primary and secondary effects of water content on leaf reflectance are greatest in spectral bands centred at 1450, 1940, and 2500 nm (Carter & Porter 1991), but has also been predicted using bands between 1100-1230 nm absorption features (Ustin et al. 1998; Asner et al. 2004). With respect to the other rock-derived nutrients, Galvez-Sola et al. (2015) also showed that near-infrared spectroscopy can constitute a feasible technique to quantify several macro and micronutrients such as N, K, Ca, Mg, Fe and Zn in citrus leaves of different leaves with coefficient of determination ($R^2$) varying between 0.53 for Mn and 0.98 for Ca, whereas B showed less accurate results with the use of spectroscopy. The regions of importance for prediction described in those studies were relatively similar to all the mineral nutrients analysed in our study, except for B that had the band between 1500 and 1900 as the best predictive region.

Some of most accurately predicted traits have no absorption features in the visible-to-near-infrared, but were instead estimated indirectly via constellation effects. Leaf mass per unit area (LMA) is consistently among the more accurately predicted traits using spectroscopy (Asner & Martin 2008; Serbin et al. 2014; Chavana-Bryant et al. 2016),  but is measured indirectly via its close coupling with water content and leaf structural traits (Asner et al. 2011b). Silicon (Si) concentrations were well-predicted by field spectroscopy, as recently reported by Smis et al. (2014). Silicon is absorbed by plants from the soil solution in the form of silicic acid ($H_4SiO_4$), being translocated to the aerial parts through xylem, and then deposited as phytoliths (Tripathi et al. 2011). Si is closely associated with phenol- or lignin-carbohydrate complexes (Inanaga et al. 1995), cellulose (Law & Exley 2011), and polysaccharide and peptidoglycans (Schwarz 1973). It seems  that spectroscopy is able to predict Si concentrations reliably because it integrates information on several of these foliar traits to make the predictions. Similarly, the relative high precisions for K, Fe and B predictions may be as strong as they are because information on several foliar traits are integrated. Unfortunately, foliar P concentrations are not closely predicted by spectroscopy.  RNA and DNA absorb in the ultraviolet (e.g. Tataurov et al. 2008) and phosphates in the longwave infrared, but there are no pronounced absorption features in the VSWIR region (Homolová et al. 2013) and covariance with other traits is weak , making constellation effects unreliable. Whilst a few spectroscopy studies have modelled P with some success, the spectral bands chosen differs among studies (Homolová et al. 2013) suggesting that constellation effects cannot be relied upon.

**4.2 Difficulties in measuring intraspecific variation by field spectroscopy and its implications for mapping functional traits**

Rock-derived nutrients lack absorption features in visible to shortwave-infrared region of the electromagnetic spectrum so cannot be measured directly by spectroscopy.  They can, nevertheless, be estimated indirectly by

virtue of the fact that element concentrations co-vary with organic molecules that do have strong absorption features ("constellation effects", see above). This paper identifies a problem with this approach: there were strong differences in rock-derived mineral nutrients between soil types, but we could not measure these because the concentrations of defence and structural traits were barely affected by soil type. We have shown many similarities between our study and those in tropical forests, demonstrating that this problem is likely to be widespread.

There are likely to be– implications of the constellation-effect problem for mapping functional traits using imaging spectroscopy. Ever larger areas of earth are being mapped with airborne spectrometers (e.g. Asner et al. 2017) and the anticipated launch of satellite-borne sensors (e.g. EnMAP; DLR 2015; Guanter et al. 2015) will soon enable vegetation and ecosystem function to be characterised at a global scale. The effectiveness of indirect prediction of traits using constellation-effect approaches will depend critically on whether soils act as a strong filter on tree species within a particular region. In the Amazonian lowlands, Asner et al. (2015) found that variation in soil P was mirrored by changes in species composition, and that P variation among species was correlated with changes in structural and defence compounds: in this instance, indirect estimation should be effective (e.g. Dana Chadwick & Asner 2016).  However, in low-diversity temperate forests, a single tree species is often found to span many different soil types and show substantial phenotypic plasticity in some traits (Oleksyn et al. 2002; Turnbull et al. 2016). The six species growing on both chalk and alluvial soils in this study are a case in point.  In these low diversity systems, it will be much more difficult to map variation using constellation effects, for the reasons explained above. Our study confirms the power of spectroscopy for predicting biochemical and structural plant traits, but we urge caution in interpreting results when species range across contrasting soil types.

**5 Conclusions**

Trees on thin chalk soils had low concentrations of N, P and K macronutrients in their leaves than trees growing on deep alluvium, but had high concentrations of several micronutrients. Phosphorus is sequestered in insoluble forms in alkaline soils. This shortage of plant available phosphorus was associated in this study with low concentrations of foliar N and low efficiency of PSII, but had no effect on structural and defensive traits.  Trait differences were far greater among species than between soil types, for all traits except foliar P. Foliar traits predicted from VSWIR reflectance spectra matched the locations of known spectral absorption features related to proteins, starch, lignin, cellulose, hemicellulose and leaf water content. Some of the most accurately predicted traits have no absorption features in the VSWIR range, and were estimated indirectly through their covariance with structural traits that do have absorption features in that spectral region ("constellation effects") including cell wall constituents. Since these structural traits did not vary with soil type, our models were unable to reliably predict intraspecific variation in rock-derived nutrients via constellation effects. Similarities between our results and those of large-scale tropical studies suggest this problem is likely to be widespread. 
[revised manuscript text omitted]

Liu, Z., Yan, H., Wang, K., Kuang, T., Zhang, J., Gui, L., An, X. and Chang, W., 2004. Crystal structure of spinach major light-harvesting complex at 2.72 Å resolution. *Nature*, 428(6980), pp.287-292.

Ma, J.F. & Yamaji, N., 2006. Silicon uptake and accumulation in higher plants. *Trends in Plant Science*, 11(8), pp.392–397.

MacGillivray, C.W., Grime, J.P. & The Integrated Screening Programme (Isp) Team, 1995. Testing predictions of the resistance and resilience of vegetation subjected to extreme events. *Functional Ecology*, 9(4), pp.640–649. Available at: http://www.jstor.org/stable/2390156.

Marschner, H., 1995. Functions of Mineral Nutrients: Macronutrients. In *Mineral Nutrition of Higher Plants*. pp. 229–312.

Marschner, M., 2012. *Mineral Nutrition of Higher Plants*, Available at: http://books.google.com/books?id=_ahKcXXQuAC&pgis=1.

Marten, G.C., Halgerson, J.L. & Cherney, J.H., 1983. Quality prediction of small grain forages by near infrared reflectance spectroscopy. *Crop Science*, 23(1), pp.94–96.

McGill, B.J. et al., 2006. Rebuilding community ecology from functional traits. *Trends in Ecology and Evolution*, 21(4), pp.178–185.

Milton, K. & Dintzis, F.R., 1981. Nitrogen-to-Protein Conversion Factors for Tropical Plant-Samples. *Biotropica*, 13(3), pp.177–181.

Misra, A. & Tyler, G., 2000. Effects of soil moisture on soil solution chemistry , biomass production , and shoot nutrients in two native grasses on a calcareous soil. *Communications in Soil Science and Plant Analysis*, 31(October 2013), pp.37–41.

Mithöfer, A. & Boland, W., 2012. Plant Defense Against Herbivores: Chemical Aspects. *Annual Review of Plant Biology*, 63, pp.431–450.

Niklas, K.J. et al., 2005. Nitrogen/phosphorus leaf stoichiometry and the scaling of plant growth. *Ecology Letters*, 8(6), pp.636–642.

Oleksyn, J. et al., 2002. Needle nutrients in geographically diverse pinus sylvestris L. populations. *Ann. For. Sci.*, 59(4), pp.1–18.

Petisco, C. et al., 2006. Near-infrared reflectance spectroscopy as a fast and non-destructive tool to predict foliar organic constituents of several woody species. *Analytical and Bioanalytical Chemistry*, 386(6), pp.1823–1833.

Pillar, V.D., Sosinski, E.E. & Lepš, J., 2003. An improved method for searching plant functional types by numerical analysis. Journal of Vegetation S*cience*, 14(3), pp.323–332. Available at: http://dx.doi.org/10.1658/1100-9233(2003)014[0323:AIMFSP]2.0.CO;2.

Porcar-Castell, A., Tyystjärvi, E., Atherton, J., van der Tol, C., Flexas, J., Pfündel, E.E., Moreno, J., Frankenberg, C. and Berry, J.A., 2014. Linking chlorophyll a fluorescence to photosynthesis for remote sensing applications: mechanisms and challenges. *Journal of experimental botany*, 65 (15), 4065-4095.

[revised manuscript text omitted]